Citation: *Molecular Systems Biology* 9:690
www.molecularsystemsbiology.com

# Dissection of a Krox20 positive feedback loop driving cell fate choices in hindbrain patterning

Yassine X Bouchoucha[1,2,3,5], Jürgen Reingruber[1,2,3,4,5], Charlotte Labalette[1,2,3], Michel A Wassef[1,2,3,6], Elodie Thierion[1,2,3], Carole Desmarquet-Trin Dinh[1,2,3], David Holcman[1,2,3,4,7,*], Pascale Gilardi-Hebenstreit[1,2,3,7] and Patrick Charnay[1,2,3,*]

[1] Ecole Normale Supérieure, IBENS, Paris, France, [2] INSERM, U1024, Paris, France, [3] CNRS, UMR 8197, Paris, France and [4] Group of Computational Biology and Applied Mathematics, IBENS, Paris, France
[5] These authors contributed equally to this work.
[6] Present address: Institut Curie, Unité de Génétique et de Biologie du Développement, 75005 Paris, France.
[7] These authors contributed equally to this work.
* Corresponding authors. P Charnay or D Holcman, Ecole Normale Supérieure; IBENS, 46 rue d'Ulm, 75005 Paris, France. Tel.: +33 1 4432 3607; Fax: +33 1 4432 3988; E-mail: patrick.charnay@ens.fr or david.holcman@ens.fr

Although feedback loops are essential in development, their molecular implementation and precise functions remain elusive. Using enhancer knockout in mice, we demonstrate that a direct, positive autoregulatory loop amplifies and maintains the expression of Krox20, a transcription factor governing vertebrate hindbrain segmentation. By combining quantitative data collected in the zebrafish with biophysical modelling that accounts for the intrinsic stochastic molecular dynamics, we dissect the loop at the molecular level. We find that it underpins a bistable switch that turns a transient input signal into cell fate commitment, as we observe in single cell analyses. The stochasticity of the activation process leads to a graded input–output response until saturation is reached. Consequently, the duration and strength of the input signal controls the size of the hindbrain segments by modulating the distribution between the two cell fates. Moreover, segment formation is buffered from severe variations in input level. Finally, the progressive extinction of *Krox20* expression involves a destabilization of the loop by repressor molecules. These mechanisms are of general significance for cell type specification and tissue patterning.
*Molecular Systems Biology* (2013) **9,** 690; published online 24 September 2013; doi:10.1038/msb.2013.46
*Subject Categories:* simulation and data analysis; development
*Keywords:* Fgf; Krox20; rhombomere; stochastic model; transcriptional enhancer

## Introduction

Cell fate specification is essential to metazoan development. It usually involves successive choices during which cells have the potential to commit to two distinct fates. Understanding the basis of cell fate specification can therefore be reduced to unravelling the molecular mechanisms underlying such choices (Graham *et al*, 2010 and references therein). Fate decisions can be induced by intrinsic cues, which may be asymmetrically distributed during cell division (Tajbakhsh *et al*, 2009; Graham *et al*, 2010), extrinsic factors that are provided by the cellular environment (Briscoe, 2009), or both. Once a choice has been made, it may become irreversible to preserve tissue integrity and directionality of the developmental process. Fate choices can be implemented by biochemical mechanisms involving feedback loops, which are maintained independently of the initial activating signal (Laslo *et al*, 2006; Graham *et al*, 2010). A single transcription factor that positively regulates its own expression constitutes the simplest genetic network that can generate a bistable switch underlying binary choices (Meinhardt, 1982; Ferrell,

2002; Tajbakhsh *et al*, 2009; Graham *et al*, 2010). The first autoregulatory transcription factor mechanism characterized experimentally was the lambda repressor, which underlies the choice between lysogeny and lytic cycle in bacteria. This example provided a paradigm for a genetic switch (Ptashne, 1986) and revealed the cooperative binding of the repressor to its DNA target as well as additional mechanisms ensuring robust efficiency. Although positive feedback loops are likely to be essential in many developmental processes, their dynamics, molecular implementation and precise functions remain elusive in vertebrates.

The establishment of hindbrain anterior–posterior (AP) identity involves a transient segmentation, which leads to the formation of seven to eight segments called rhombomeres (r) (Lumsden and Krumlauf, 1996; Briscoe, 2009). Rhombomeres constitute cell compartments and developmental units for neuronal differentiation and branchiomotor nerve organization (Lumsden and Keynes, 1989). The transcription factor Krox20 (also known as Egr2) is specifically expressed in r3 and r5, and is required for the formation and specification of these

rhombomeres, in particular for establishing odd- versus even-numbered identity (Schneider-Maunoury *et al*, 1993; 1997; Swiatek and Gridley, 1993; Voiculescu *et al*, 2001). Three evolutionarily conserved transcriptional enhancer elements active in the hindbrain have been identified in the *Krox20* locus, termed A, B and C (Chomette *et al*, 2006; Wassef *et al*, 2008). Elements B and C drive the expression of reporter constructs in r5 and r3/5, respectively. Their activity is independent of Krox20, but is modulated by FGF signalling (Labalette *et al*, 2011). They are thought to be involved in the initiation of *Krox20* expression. Element A drives reporter expression in r3 and r5, and contains several Krox20-binding sites. The integrity of these sites are necessary for its activity, suggesting that element A is involved in a positive feedback loop (Chomette *et al*, 2006).

In the present study, to understand the mechanisms of a vertebrate autoregulatory loop, we took advantage of our extended knowledge of *Krox20* regulation and of the high conservation during vertebrate evolution of the molecular mechanisms governing its expression, including the activities of the *cis*-acting elements. This conservation allowed us to move between species (mouse, chick or zebrafish), taking advantage of the specificities of each experimental system. Using a knockout approach in the mouse, we established that element A is indeed required for Krox20 positive feedback in the developing hindbrain. As Krox20 DNA-binding, promoter activation, transcription, translation and degradation events are intrinsically subject to fluctuations, we developed a stochastic model to quantify and predict the role of these fluctuations in the determination of cell fate. Because of these fluctuations, a fixed input signal leads to variability in the cell fate choice. Using our stochastic model, we determine the bimodal cell fate distribution that a transient input signal induces into a homogenous population of cells. By combining quantitative data collected in the zebrafish with stochastic modelling, mathematical analysis and numerical simulations, we reach an unprecedented understanding of the molecular dynamics underlying a vertebrate patterning process at the cellular level.

## Results

### Element A controls Krox20 autoregulation in the mouse hindbrain

Element A activity is dependent on the direct binding of Krox20, shown in mouse and chick (Chomette *et al*, 2006), raising the possibility that this element is responsible for the autoregulation of the gene in the hindbrain. To establish this point, we generated a knockout of element A in the mouse. The details of the strategy are presented in Figure 1A. Two alleles were generated: $Krox20^{\Delta A}$, where element A is deleted and $Krox20^{A*}$, where element A is replaced by element cA*. cA* is a version of the chick element A that contains specific mutations in the Krox20-binding sites that prevent binding of

wild-type (WT) Krox20 and instead allow binding of a mutant Krox20 protein, Krox20* (Nardelli *et al*, 1991; Supplementary Figure S1A–C).

The consequences of element A deletion on *Krox20* expression were analysed by mRNA *in-situ* hybridization in homozygous mutant embryos (Figure 1B). No differences are observed between $Krox20^{\Delta A/\Delta A}$ embryos and their littermate $Krox20^{\Delta A/+}$ or WT controls until approximately six somites (s). At 6s, the level of *Krox20* mRNA and the extension of the r3 domain are slightly reduced in the homozygous mutants compared with that in controls (Figure 1B). At 8s, r3 expression is almost lost in $Krox20^{\Delta A/\Delta A}$ embryos, whereas it persists beyond 12s in controls. In r5, a reduction in the level of *Krox20* mRNA compared with that in controls is observed from 8s and expression is lost around 12s, whereas it persists beyond 16s in controls. These data indicate that the 465-bp sequence deleted in the mutant is required for both amplification and maintenance of *Krox20* expression and that element A is a key component of the *Krox20* autoregulatory loop.

To investigate the consequences of this altered *Krox20* expression on hindbrain patterning, we analysed the expression of one of the Krox20 target genes, encoding the tyrosine kinase receptor EphA4 (Theil *et al*, 1998). *EphA4* is normally expressed at high relative levels in r3 and r5, and at low levels in r2 (Figure 1C; (Gilardi-Hebenstreit *et al*, 1992)). *EphA4* expression persists in r3 and r5 after *Krox20* is switched off (unpublished observations), indicating that at some point it becomes independent of Krox20 and is therefore a marker of commitment to the r3/r5 fate. In $Krox20^{\Delta A/\Delta A}$ embryos, the size of the domains of high *EphA4* expression is markedly reduced after 8s (Figure 1C). This suggests that transient expression of *Krox20* is sufficient to drive a limited number of cells into an r3/r5 fate, but that the Krox20 autoregulatory loop is required for obtaining odd-numbered rhombomeres of normal size. This observation was confirmed by direct analysis of the r2 and r4 territories, using an alkaline phosphatase reporter transgene specifically expressed in r2 (Studer *et al*, 1996) and *in-situ* hybridization against *Hoxb1* to reveal r4 (Figure 1E). This analysis shows the persistence of a reduced r3 territory at embryonic day 9. We also observed an increase in r4 size (Figure 1E), suggesting that lack of Krox20 autoregulation may lead to re-specification of cells normally fated to belong to odd-numbered rhombomeres.

To demonstrate that the $Krox20^{\Delta A/\Delta A}$ phenotype specifically originates from a defect in *Krox20* autoregulation, we attempted a rescue by re-establishing a positive feedback loop based on the specific interaction between Krox20* and cA* (Supplementary Figure S1D–F). We verified that $Krox20^{A*/A*}$ and $Krox20^{\Delta A/\Delta A}$ embryos display similar phenotypes (Figure 1D). We then generated a transgenic mouse line, *Tg(cA:Krox20*)*, carrying the *Krox20** coding sequence under the control of the chick element A, which responds to mouse Krox20 and is active in r3 and r5 (Chomette *et al*, 2006). In *Tg(cA:Krox20*);Krox20^{A*/A*}* embryos, hindbrain expression

**Figure 1** Element A is required for the maintenance of mouse *Krox20* expression. (**A**) Knockin into the A element and deletion strategy. The different alleles obtained after homologous Cre (targeting loxP sites) or Flp (targeting FRT and F3 sites) recombination are presented. cA* is a mutant form of chick element A that can only be bound by a mutant version of Krox20, termed Krox20* (see Supplementary Figure S1A–C). *Krox20* (**B**) and *EphA4* (**C**) *in-situ* hybridization performed on $Krox20^{\Delta A/+}$ (control) and $Krox20^{\Delta A/\Delta A}$ embryos at the indicated stages. (**D**) *EphA4 in-situ* hybridization performed on $Krox20^{A*/A*}$ embryos carrying or not the *Tg(cA:Krox20*)* transgene. (**E**) *Hoxb1 in-situ* hybridization (revealing r4) and alkaline phosphatase staining (revealing r2) performed on $Krox20^{A*/A*}$; *r2-HPAP* embryos at day 9 of embryonic development (25s approximately).

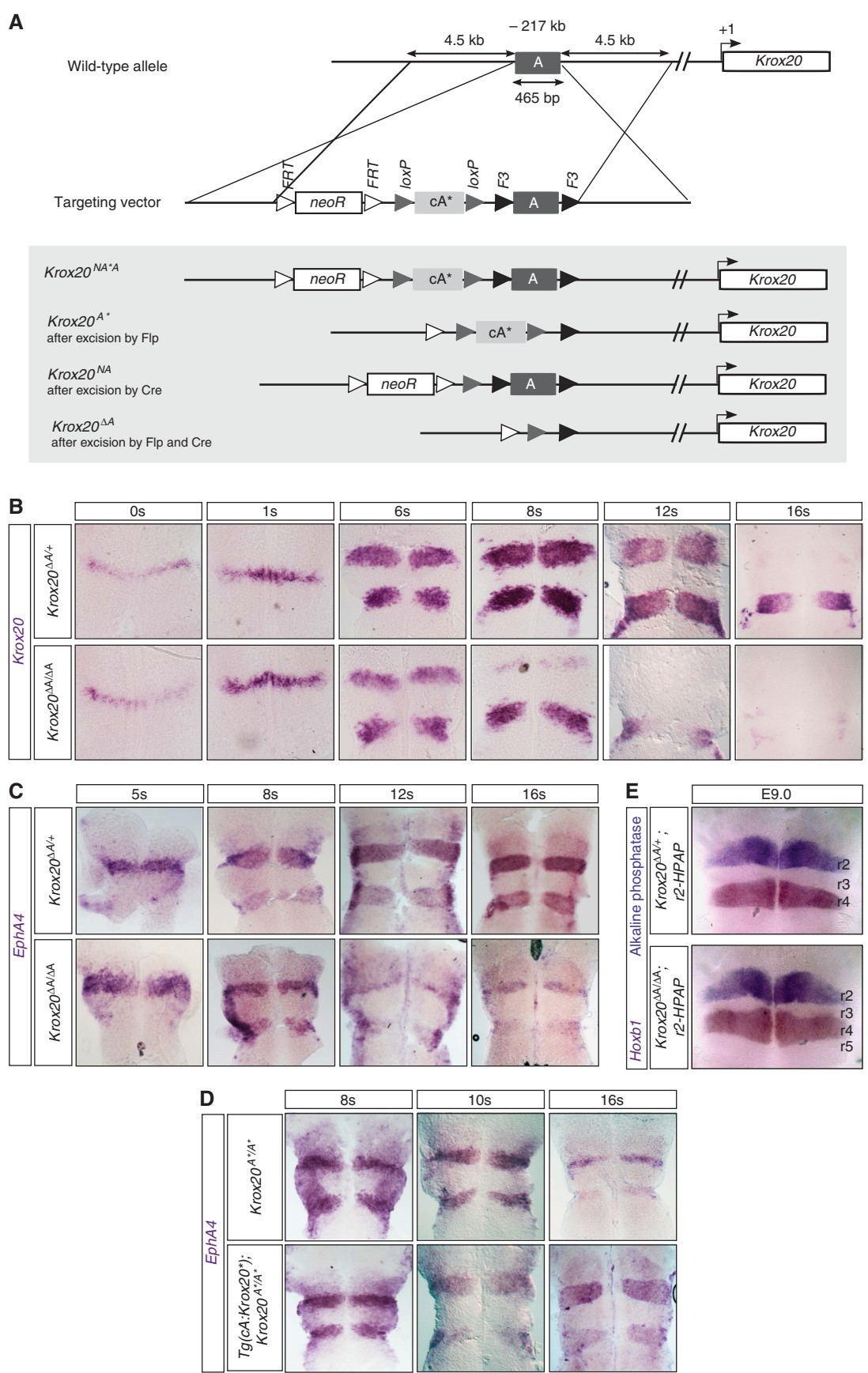

of *EphA4* was largely rescued (Figure 1D). In this situation, Krox20 is normally initiated from the endogenous locus and activates the exogenous element A, driving the expression of Krox20*; in turn, Krox20* activates element A* on the endogenous locus, leading to further production of Krox20 (Supplementary Figure S1F). The damaged, endogenous loop is rescued by a novel, indirect autoregulatory loop. This analysis demonstrates that the phenotype associated with the $Krox20^{\Delta A/\Delta A}$ mutation is exclusively due to a lack of *Krox20* autoregulation. Furthermore, our data establish that the activation of element A involves direct binding of Krox20, as mutations in the *cis*-acting element can be rescued by compensatory mutations in the *trans*-acting factor.

## Krox20 autoregulation is conserved in zebrafish

To quantitatively analyse the features of Krox20 autoregulation, we used the zebrafish embryo. We first investigated the existence of an autoregulatory loop in this species. A fish line carrying a point mutation in the *krox20* coding sequence that abolishes Krox20 function (*krox20^fh227* allele (Monk *et al*, 2009)) and, therefore, autoregulation was used to perform a loss-of-function analysis. The mutation does not prevent activation of *krox20* expression in r3 and r5, but rapidly leads to its extinction from 6s in r3 and 10s in r5, whereas expression is maintained in both rhombomeres beyond 14s in WT embryos (Figure 2A–F and A′–F′). This phenotype indicates that Krox20 is required for maintaining its own expression. We then engineered a transgenic fish line, *Tg(hsp:mKrox20_HA)*, in which a HA-tagged

murine version of *Krox20* is under the control of a heat-shock (HS) promoter. Using this system, we can modulate the amount of *mKrox20* produced in all embryonic cells by modifying the temperature or duration of the HS. HSs performed at increasing temperatures lead to progressive activation of endogenous *zkrox20* in r2, r4 and r6 (Figure 2G–J). Although expression of the transgene occurs in the entire embryo, efficient activation of the endogenous gene is essentially restricted to the hindbrain, presumably due to additional and unknown regulatory mechanisms. Together, these data establish the existence of a Krox20 autoregulatory loop in the zebrafish.

We were unable to find a fish orthologue of element A by an *in-silico* search based on nucleotide conservation. However, we identified a 1-kb DNA fragment that can drive specific expression of a reporter gene in r3 and r5 in the zebrafish (Supplementary Figure S2). This fragment contains a cluster of five potential Krox20-binding sites, is located upstream of the *krox20* gene, at a position corresponding approximately to that of the chick element A, and has been shown to co-immunoprecipitate with the H3K4m1 histone modification, which usually marks the enhancers (Aday *et al*, 2011). It is therefore likely to contain zebrafish element A.

## Analysis of the molecular mechanisms governing *Krox20* expression

To evaluate the respective contributions of initiation and autoregulation to *krox20* expression, we measured *krox20* mRNA levels in WT and *krox20^fh227/fh227* zebrafish embryos by

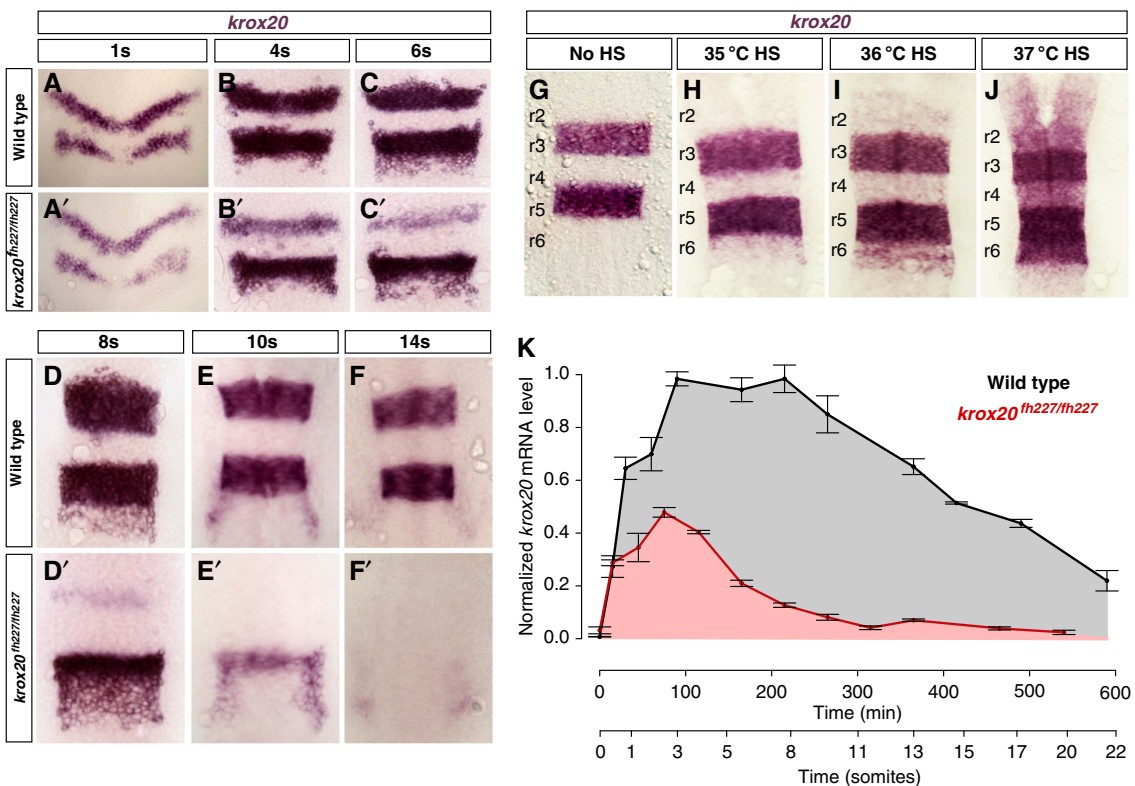

**Figure 2** *Krox20* autoregulation in the zebrafish embryo. (**A–F** and **A′–F′**) *krox20 in-situ* hybridization performed on WT and *krox20^fh227/fh227* embryos at the indicated stages. (**G–J**) *krox20 in-situ* hybridization on 15s *Tg(hsp:mKrox20_HA)* embryos after 10 min HS performed at 1s at the indicated temperatures. Rhombomeres (r) are indicated. (**K**) Time course of *krox20* mRNA level measured by RT–qPCR in WT (black curve) and *krox20^fh227/fh227* (red curve) embryos. The data are normalized with the value of the WT plateau. Source data for this figure is available on the online supplementary information page.

reverse transcriptase–quantitative PCR (RT–qPCR), between 100% epiboly and 22s (Figure 2K). During this period, *krox20* expression is restricted to the hindbrain and the measurements therefore correspond to the added levels of r3 and r5. Mutant expression reflects only the initiation process, whereas WT expression corresponds to the combined output of initiation and autoregulation. In the mutants, *krox20* mRNA rapidly accumulates, peaks around 3 s and then decays. The presence of the autoregulatory loop leads to a twofold increase in the maximal mRNA level and to an extension of the expression period, with a plateau between 3s and 8s, followed by a linear-like decline. Therefore, the initiator elements provide only a short pulse of *krox20* expression that is necessary to trigger the autoregulatory loop, which results in 4.7-fold higher dose of

*krox20* mRNA during the 0–22s period (compare areas under the curves in Figure 2K).

Transcription systems often rely on cooperative transcription factor binding to DNA and on synergistic activation (also known as concerted recruitment) of the transcriptional machinery by multiple transcription factors (Georges *et al*, 2010). We investigated whether element A activity involves such mechanisms. *In-silico* analysis of the 416-bp chick element A revealed the presence of seven putative Krox20-binding sites (Figure 3A and (Chomette *et al*, 2006)). *In-vitro* binding of Krox20 to each of the sites was assessed by competitive electromobility shift assay (EMSA). Three of the sites were of high affinity and one of medium affinity (sites K2, K5, K7 and K6, respectively; Supplementary Figure S3). The

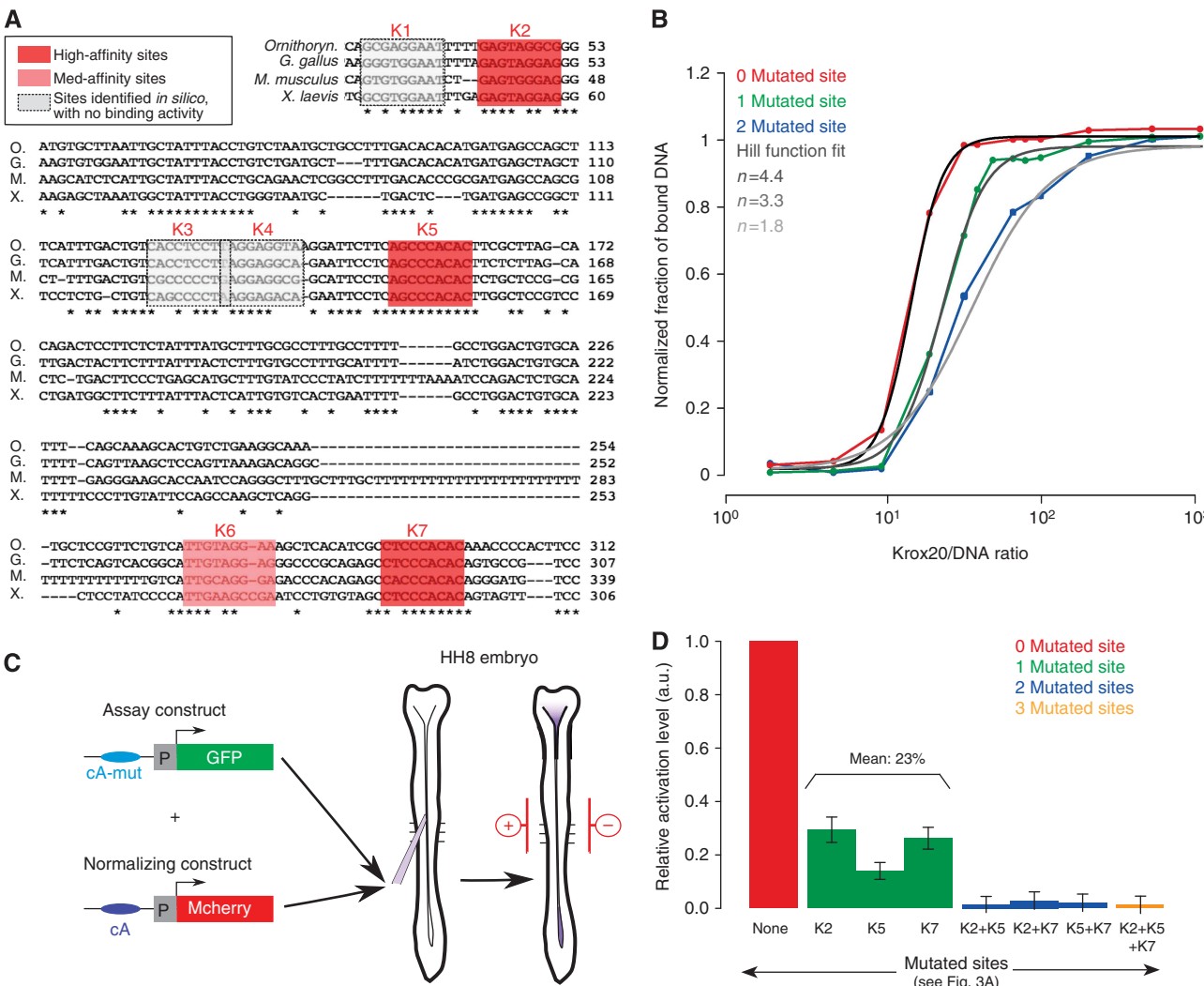

**Figure 3** Binding cooperativity and synergy in transcriptional activation. (**A**) Alignment of element A nucleotide sequences from different vertebrate genomes (O, *Ornithorynchus*; G, *Gallus gallus*; M, *Mus musculus*; X, *Xenopus laevis*). Stars indicate conserved nucleotides. Bona fide Krox20-binding sites in the chick enhancer are highlighted in red or pink. (**B**) Saturation curves of Krox20 binding on chick element A obtained by EMSA experiments. The experiment was performed on WT element A (red) and mutant variants carrying either one mutated Krox20 site (site K2, green) or two mutated sites (sites K2 and K5, blue). Fitting with a Hill function is shown (grey curves). (**C**) Experimental design of the chick electroporation assay. Two constructs were co-electroporated in the chick embryo neural tube at stage HH8: the assay construct, where WT or mutated element A drives *GFP* expression, and the normalization construct, where WT element A drives *mcherry* expression. In all experiments, the level of GFP was normalized with the level of mCherry fluorescence. (**D**) Quantification of the electroporation experiments. GFP levels of the different element A mutants (green, blue, orange columns) are displayed relative to WT (red column). Source data for this figure is available on the online supplementary information page.

other putative sites had very low or no binding activity and were no longer considered. We introduced deleterious mutations into the three high-affinity Krox20-binding sites, either alone or in combination, and performed saturation experiments, where a fixed concentration of element A is incubated with increasing amounts of protein (Figure 3B). The data demonstrate strong cooperative binding. Fitting each curve with a Hill function (Figure 3B) provided Hill coefficients of 4.4 for WT element A, 3.2 and 3.3 for single mutants and 1.8 for the double mutant. These data are consistent with cooperative binding involving each of the four sites (Ma *et al*, 1996; Burz *et al*, 1998).

To evaluate synergy in transcriptional activation, we measured the relative activity of each element A mutant in a co-electroporation assay in the chick hindbrain, which allows an easy and quantitative comparison of the steady-state dynamics in the hindbrain of exogenous enhancer elements driving the expression of reporter genes (Chomette *et al*, 2006; Wassef *et al*, 2008). In this assay, expression of a reporter driven by element A is essentially restricted to r3 and r5 (Chomette *et al*, 2006). Two reporter plasmids were co-electroporated: a construct in which WT or mutant element A drives GFP expression and a normalization construct, in which mcherry is driven by WT element A (Figure 3C). Eighteen hours after electroporation, for each mutant the normalized fluorescence level associated with GFP in r3 and r5, relative to WT, was taken as a measure of the relative activity of the element. Single mutations reduce element A transcriptional activity to approximately one-fourth of the control, whereas combinations of two or three mutations abolish it completely (Figure 3D). The non-additive contributions of the binding sites demonstrate that Krox20 molecules bound to the enhancer activate transcription in a synergistic manner.

## A stochastic model for *Krox20* transcriptional regulation

To study the mechanisms governing *Krox20* expression, we developed a stochastic mathematical model based on molecular dynamics. We modelled the dynamics of Krox20 mRNA and proteins, cooperative binding/unbinding of Krox20 proteins to the four binding sites of element A and synergy for transcriptional activation. We also modelled the transient initiation phase that allows activation of *Krox20* expression. As shown below, only a few mRNA molecules are involved in this activation and a stochastic approach was required to correctly account for large fluctuations.

The model is schematically represented in Figure 4A and is fully described in the Supplementary Information. *Krox20* expression is initially zero. From time $t = 0$ to $t_I$, in addition to autoregulation, an initiation process leads to the production of *Krox20* mRNA by a mechanism that is independent of autoregulation. This mechanism can be either endogenous (through the activity of initiator *cis*-acting elements), or exogenous (upon HS in the $Tg(hsp:mKrox20_{HA})$ line). *Krox20* mRNA is produced by this process at a Poissonian rate $\Phi_I$. This Krox20-independent initiation process is responsible for an initial production of Krox20 that can binds to element A and activates the autoregulatory loop. Krox20 protein production

due to a single mRNA occurs at a Poissonian translation rate $\phi$. mRNA molecules are degraded with a Poissonian rate $\Psi$ and proteins with a rate $\psi$. Proteins bind to element A, thereby also activating and modulating mRNA production. Element A is modelled with four equivalent binding sites and can be in five states ($s = 0, 1, 2, 3, 4$), depending on the number of bound Krox20 proteins. We implemented cooperativity in unbinding from element A using the state-dependent unbinding rates $\mu_s = \mu/\gamma_s$, where $\mu$ is an overall rate constant and the numbers $\gamma_s$ describe the modulations due to the state of element A (see section 2.8 in Supplementary Information). In contrast, we assumed that binding is not affected by the state of element A and used a single forward binding rate $\lambda$. We implemented transcriptional synergy in element A activity by using a state-dependent mRNA production rate $\Phi_{A,s} = \Phi_A \xi_s$, where $\Phi_A$ is the maximal production rate when all four sites are bound and the numbers $\xi_s$ describe the modulation by the state of element A. Finally, the model takes into account the two *Krox20* alleles, the parameters $\Phi_I$ and $\Phi_{A,s}$ corresponding to a single allele.

The model is characterized by three dynamic variables: the number $m$ of *Krox20* mRNA molecules, the number $n$ of proteins and the state $s$ of element A. To compute these variables, we derived the Master equations (equation (1) in Supplementary Information) to obtain the joint probability $p_s(m,n,t)$ to find $m$ mRNA molecules, $n$ proteins and element A in state $s$ at time t. To separate initiation from autoregulation, we distinguished the numbers of Krox20 mRNAs and proteins produced by initiation, Krox20[I], and by element A, Krox20[A]. The model depends on 14 parameters listed in Supplementary Table S1. Section 2 of the Supplementary Information describes how 10 of these parameters were determined experimentally and 2 derived from the literature. To obtain the remaining two parameters, we developed an indirect parameterization approach, where we compared simulations with experimental data.

This approach allowed us to derive both the dynamics of *Krox20* expression and the fluctuations from elementary molecular events. The fraction of cells that eventually activate element A depends on the initiation period and therefore cannot be computed from steady-state analysis. To study the time evolution of the system, we numerically solved the Master equation to obtain $p_s(m,n,t)$, from which we computed all statistical properties. In addition, to study the evolution of a single cell, we used the Gillespie algorithm (Gillespie, 1976; Tajbakhsh *et al*, 2009; Graham *et al*, 2010) to simulate the molecular reactions underlying the Master equations. We refer to these single cell traces as 'molecular dynamics simulations'.

## The model accounts for the dynamics of *krox20* activation

We first implemented the model to study the dynamics of *krox20* expression. In Figure 4B, we compare the measured time course of *krox20* mRNA expression in WT and *krox20*[fh227/fh227] embryos (black curves) with numerical simulations (red curves). In the WT, the activation of element A is saturating (see part 2.3 of Supplementary Information) and we used the plateau value as a reference for the maximal mRNA level to normalize the experimental data (Figure 2K).

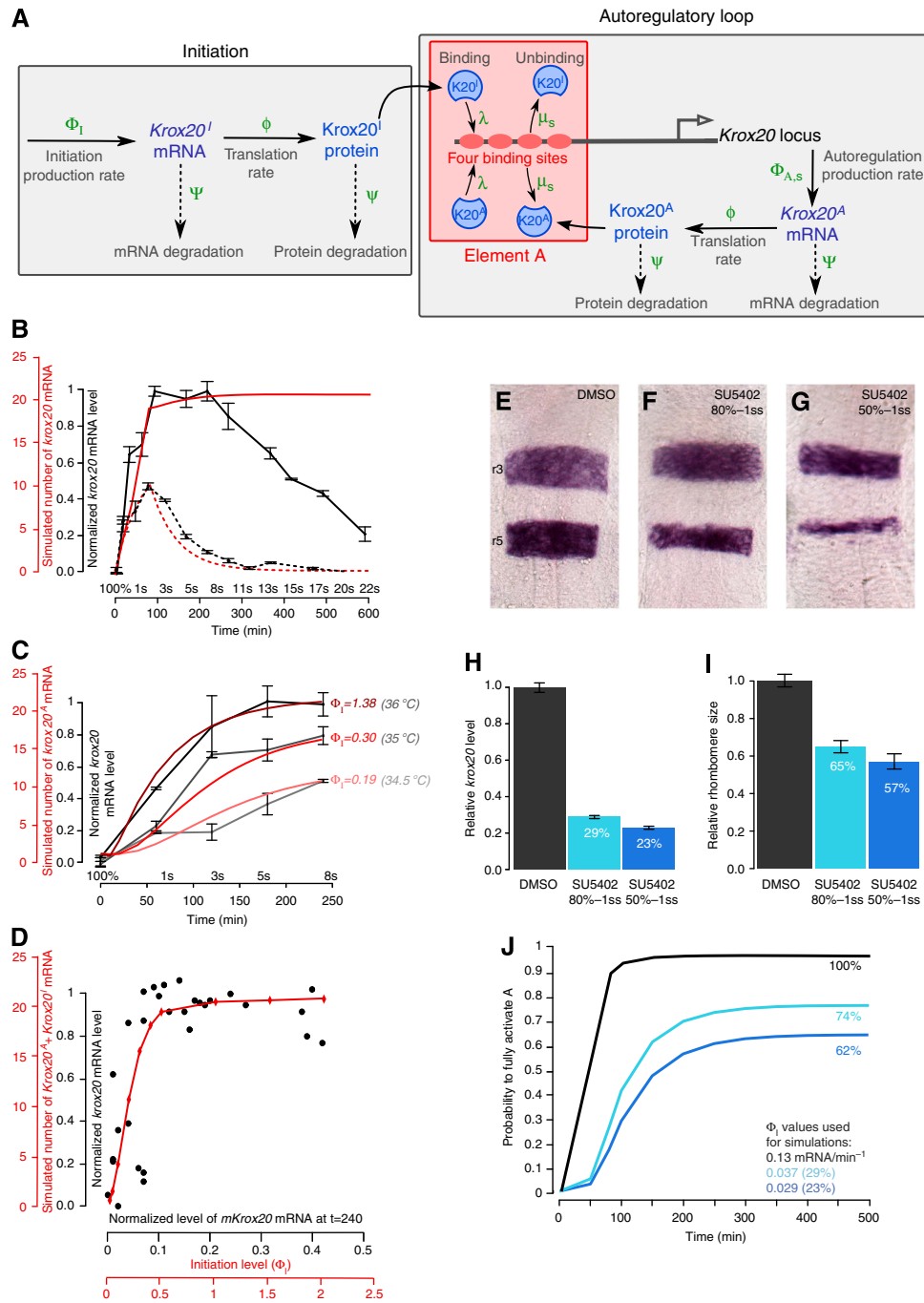

**Figure 4** A quantitative model of *Krox20* expression. (**A**) Schematic representation of the molecular events used to model *Krox20* expression. *Krox20* expression proceeds in two phases: during a first phase that lasts for $t_i$ minutes, Krox20 production involves both a Krox20-independent initiation process and the autoregulatory loop, whereas only autoregulation is maintained afterwards. For further details, see the text. (**B**) Comparison of simulations (red) and experimental data (black) of *Krox20* expression time courses in developing embryos with (solid curves) and without (dashed curves) activity of the autoregulatory loop. For the simulations, we used 80 min initiation time and rates $\Phi_I \approx 0.7\Phi_A = 0.13$ mRNA/min. (**C**) Time courses of *krox20* expression obtained from experimental data (grey curves) and simulations (red curves) following HS at the indicated temperatures. To obtain the experimental level of *krox20* mRNA present in even-numbered rhombomeres, we subtracted the level of *krox20* mRNA present in non-transgenic fish, as determined in **B**. The HS simulations were performed with 10 min initiation time and the indicated initiation rates. (**D**) Experimental (black dots) and simulated (red curve) input/output relationship of element A activation extracted from the HS data. Each point (black dot) corresponds to the levels of *mKrox20* and *zkrox20* RNA measured by RT–qPCR 4 h after HS in a single embryo. Embryos were subjected to 10 min HS ranging from 33 to 39 °C. (**E–G**) *krox20 in-situ* hybridization on 15s WT zebrafish embryos treated under two conditions with the FGFR inhibitor SU5402 (**F**, **G**) or mock-treated (**E**). (**H**, **I**) Diagrams showing the relative *krox20* mRNA levels in *krox20*[fh227/fh227] embryos measured by RT–qPCR (**H**) and the relative area of r3 + r5 in WT embryos treated under the two SU5402 conditions compared with mock treatment (DMSO). (**J**) Simulations of the probability to fully activate A ($p_4$) when the initiation rate is altered. The simulations were performed with 80 min initiation time, and the black curve (labelled 100%) was obtained with the control rates $\Phi_I = 0.13$ mRNA/min. For the other two conditions, $\Phi_I$ was reduced to 29 and 23% of the control value, corresponding to the experimental measurements in H. The steady-state level for $\Phi_I = 0.13$ mRNA/min is defined as 100%. Source data for this figure is available on the online supplementary information page.

Figure 4B was obtained from numerical simulations of the joint probability $p_s(m,n,t)$ (equation 1 in the Supplementary Information), from which we computed the time course of the mean number of mRNA per cell. To compare experiments with simulations, we scaled the number of mRNA/cell with the steady-state number of mRNA/cell provided by a fully activated element A. For the WT simulations, we used two alleles per cell; we estimated $\Phi_I \approx 0.7\Phi_A = 0.13$ mRNA/min for each allele, and we assumed that initiation in r3 and r5 was the same and lasted for 80 min.

We found that the model accounted for the dynamics *krox20* expression in the mutant and its upregulation and early plateau phases in the WT (Figure 4B). However, it did not account for the decrease observed after 8s in the WT. This difference will be discussed in detail below.

Our experimental measurements of *krox20* mRNA levels by RT–qPCR were obtained from whole embryos and, therefore, correspond to the summation of the levels in r3 and r5. The dynamics of *krox20* expression in the absence of autoregulation are shifted in time between r3 and r5 (Figure 2A′–F′). Therefore, our model with a single input represents a simplification of this situation. However, we have performed numerical simulations based on our model with two shifted inputs of the same $\Phi_I$ value for r3 and r5, and the differences with simulations performed with a single input were marginal (Supplementary Figure S4). For simplicity, we therefore performed all following simulations with a single input.

To investigate *Krox20* expression in non-saturating conditions, we analysed the *Tg(hsp:mKrox20$_{HA}$)* transgenic line before the decline of endogenous *zkrox20* expression. In this case, the level of input *mKrox20* mRNA can be tuned experimentally by varying the HS temperature and can be discriminated from the level of *zkrox20* mRNA by RT–qPCR. We performed 10-min HSs at three temperatures (34.5, 35 and 36 °C) at 100% epiboly and measured the level of *mKrox20* mRNA at time t = 0 (end of the HS) and the level of *zkrox20* mRNA in even-numbered rhombomeres from t = 0 to 240 min (Figure 4C, grey curves). Experimental mRNA levels were normalized with the saturating level obtained at 36 °C. We next compared the experimental data with model predictions where we varied the initiation production rate $\Phi_I$. As the experimental initiation rates $\Phi_I$ are not known, we first estimated one reference value for $\Phi_I$ (0.19 mRNA/min) by fitting the experimental curve at 34.5 °C (Figure 4C). The $\Phi_I$ values at 35 °C (0.30 mRNA/min) and 36 °C (1.38 mRNA/min) were computed such that the ratios of $\Phi_I$ values equal the ratios of experimental input values. Both simulations obtained with these calculated $\Phi_I$ values were also in agreement with the experimental data (Figure 4C), showing that the model correctly predicts *krox20* expression in both saturating (36 °C) and non-saturating conditions (34.5 °C and 35 °C).

Finally, we used the HS data to extract a dose–response curve giving the levels of *zkrox20* mRNA as a function of the levels of *mKrox20* mRNA (Figure 4D, black dots). Both mRNA levels were measured 240 min after HS, i.e., close to the steady state according to the kinetics shown in Figure 4C. A simulated dose–response curve (in red in Figure 4D) was obtained by plotting the number of *Krox20$^A$* mRNA molecules as a function of the initiation rate $\Phi_I$. The agreement between the simulation

and the experimental data shows that the model correctly predicts the input–output relationship of this system.

## The feedback loop underpins a bistable switch and induces a bimodal cell distribution

We used the model to analyse the autoregulatory loop. We first investigated in which conditions positive feedback leads to a bistable switch. We found that a stable state with a high *Krox20* expression level exists when the product of the production rate $\Phi_A$ and the ratio $\beta = \lambda/\mu$ exceeds a threshold value (Supplementary Figure S5 and S6 and section 3.2 of Supplementary Information). With one or two alleles and a production rate $\Phi_A = 0.18$/min (Supplementary Information, section 2.4), this results in the minimal values $\beta_{min} = 0.13$ or $\beta_{min} = 0.065$, respectively. We estimated a value of 0.20 for $\beta$ (Supplementary Information, section 3.2.2), which is largely above the minimal values. Therefore, stable maintenance of the loop is guaranteed in both WT and heterozygote.

We then studied how the initiation phase affects *Krox20* expression. We find that the fraction of the cells that commit to a Krox20-positive fate can be modulated by varying either the initiation strength $\Phi_I$ or its duration $t_I$ (Supplementary Figure S7). Furthermore, the gradual splitting of the original homogeneous cell population leads to two different homogeneous populations, as the stochastic properties of the initiation process do not introduce additional variability (Supplementary Figure S7D, H). We illustrate these findings in Figure 5 by dissecting the dynamics of *Krox20* activation for values of $\Phi_I$ corresponding to the HS temperature conditions presented in Figure 4C. The results presented in Figure 5A–F were obtained from numerical simulations of the joint probability $p_s(m,n,t)$ (equation 1 in the Supplementary Information). The time course of the probability $p_s(t)$ of finding element A in state *s* (referred to as 'state probability', Figure 5A–C) reveals that element A is either fully activated (s = 4, four Krox20 proteins bound) or fully deactivated (s = 0, no protein bound) and the probability of the intermediate states are negligible. As a consequence, the steady state value of $p_4(t)$ for large time periods yields the fraction of cells that select a Krox20-positive fate. Figure 5A shows that an average initial production of about four mRNA molecules per cell (0.19 mRNA/min/allele × 10 min × 2 alleles ≈ 4 mRNA) induces the Krox20-positive fate with 60% probability (see also Supplementary Figure S7I). Thus, the critical region that determines cell fate involves the action of only a few mRNA molecules, leading to a highly stochastic regime. The heat maps of the time-dependent probability for the number of Krox20 proteins within a cell show two strands corresponding to the bimodal distribution of Krox20-positive versus Krox20-negative cells (Figure 5D–F). Increasing the initiation rate $\Phi_I$ changes the occupancy of the large strand, but not its location nor width, demonstrating that the initiation does not affect the characteristics of Krox20-positive cells. For large $\Phi_I$, almost all cells express *Krox20* and the probability distribution is essentially unimodal (Figure 5F). Figure 5G–I show the time evolution of *Krox20* expression in individual cells, obtained from the molecular dynamics simulations. Cells either evolve toward a stable high-level *Krox20* expression or the expression vanishes. For cells that have reached high-level expression, this state is

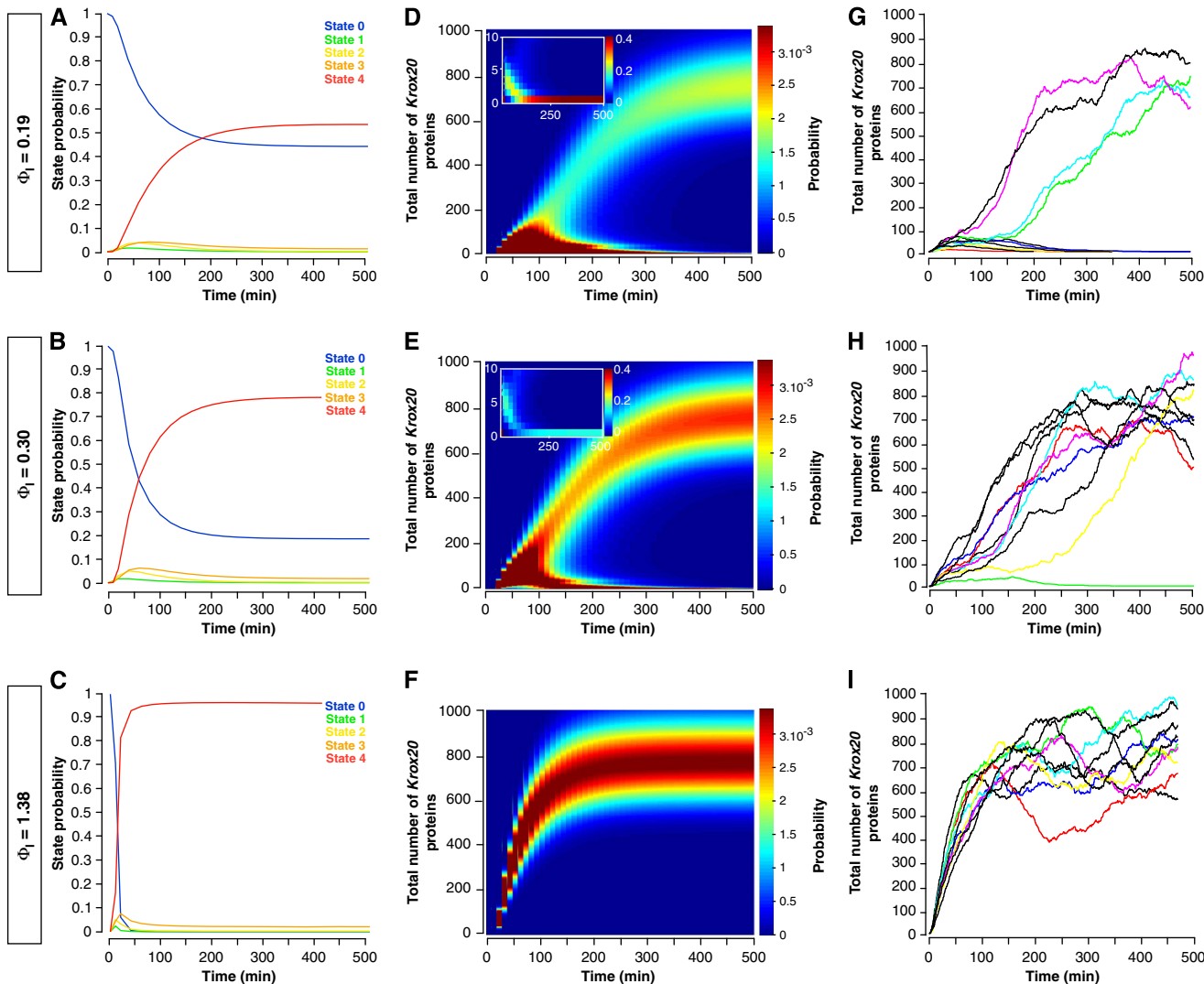

**Figure 5** Dynamics of *Krox20* activation derived from numerical simulations. (**A–C**) Time-dependent probability of the different states of element A ($s = 0, 1, 2, 3$ or $4$) for three different levels of initiation ($\Phi_I$), corresponding to the HS conditions presented in Figure 4C with 10 min initiation time. (**D–F**) Heat maps for the evolution of the probability distribution for the total number of Krox20 proteins in a cell with initiation as in **A–C**. Insets are zooms for low protein numbers. (**G–I**) Molecular dynamics simulations for the evolution of the stochastic number of Krox20 proteins in single cells with initiation as in **A–C**. Each panel shows the time course in 10 cells. The results shown in panels **A–F** were obtained by integrating the Master equation 1 in the Supplementary Information. The molecular dynamics simulations shown in **G–I** are obtained from the Master equation 1, using the Gillespie algorithm.

maintained. In the critical region of initiation values, the evolution is highly stochastic, leading to a large variability in the transition time to the final state (Figure 5G and H).

Finally, we explored how changes in binding cooperativity and synergistic promoter activation affect the bistable behaviour of the system. We find that both are required to obtain the following properties (see section 3.2.5 of Supplementary Information and Supplementary Figure S8): (1) efficient activation of the loop; (2) existence of a bimodal state with either fully activated or fully deactivated promoter; (3) the fraction of cells that commit to a Krox20-positive fate is sensitive to and gradually modulated by low initiation levels.

In conclusion, our model shows that an initial fluctuating Krox20 signal is converted by the autoregulatory loop into a bistable behaviour conditioning cell fate choice, and the distribution between the two fates is determined by the strength of the initiation signal and depends on cooperative binding and synergistic production.

## Experimental demonstration of bimodality

Bimodality is predicted by the model for low-level inputs. To verify this experimentally, we used a zebrafish transgenic line carrying a reporter construct in which a histone h2b-mcherry fusion protein is placed under the control of chick element A, *Tg(cA:h2b-mcherry)*. In this condition, *mcherry* is expressed in r3 and r5, and localizes in the cell nucleus. This reporter line was crossed with the *Tg(hsp:mKrox20$_{HA}$)* line (Figure 6A). As expected, in the absence of HS, we observe specific and homogeneous nuclear mCherry fluorescence in r3 and r5 cells, in about half of the embryos (Figure 6B and C). These positive

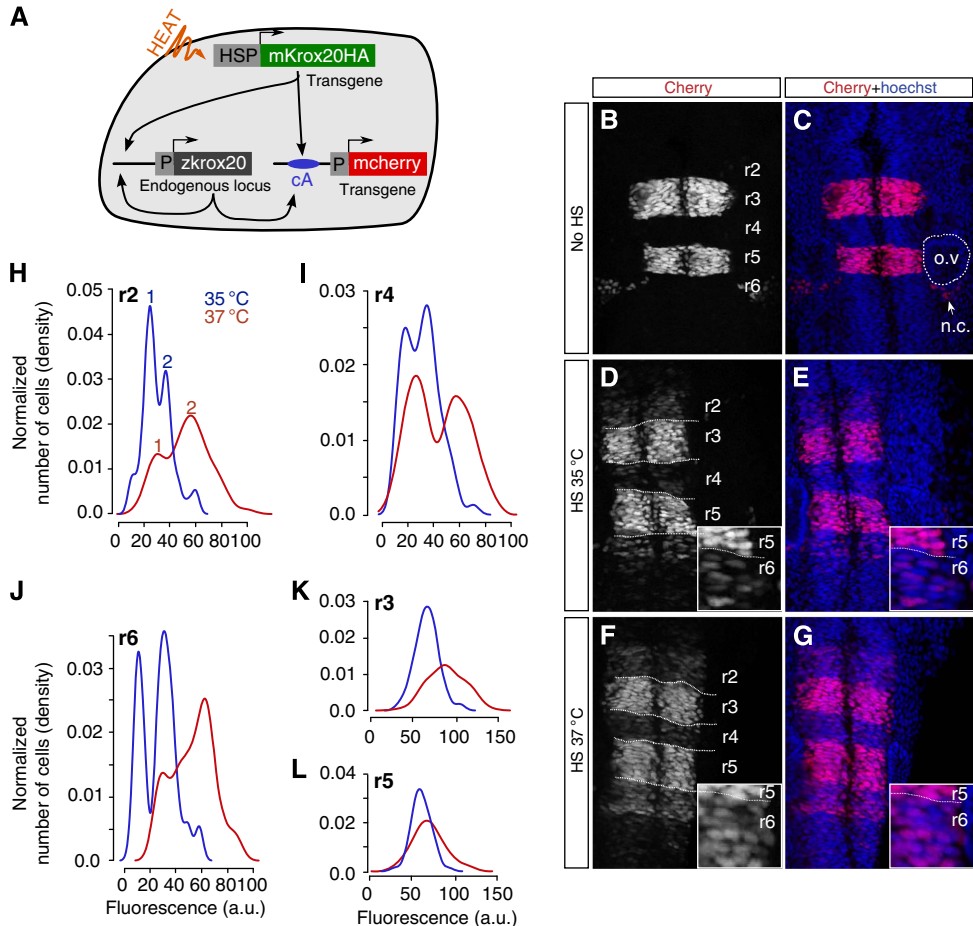

**Figure 6** Experimental manifestation of bimodality. (**A**) Schematic representation of the system designed to measure element A activity at the single-cell level. HSs in double transgenic *Tg(hsp:mKrox20_HA);Tg(cA:h2b-mcherry)* embryos produced an ectopic activation of element A. This activation was quantified by measuring nuclear mCherry fluorescence in single cells. (**B**–**G**) Confocal pictures showing mCherry and Hoechst 33342 fluorescence in double transgenic embryos that did not undergo HS (**B**,**C**) or were HS treated for 10 min at 35 °C (**D**,**E**) or 37 °C (**F**,**G**) at 5s. (**H**–**L**) Distribution of the number of cells according to their level of mCherry fluorescence in r2 (**H**), r4 (**I**), r6 (**J**), r3 (**K**) and r5 (**L**) measured in the 35 °C (blue curve) and 37 °C (red curve) HS experiments; o.v., otic vesicle; n.c., neural crest.

embryos correspond to those having inherited the *Tg(cA:h2b-mcherry)* transgene. Approximately 25% of the HS embryos (35 °C or 37 °C) showed mCherry fluorescence 4 h later in even-numbered rhombomeres (Figure 6D–G). They correspond to embryos having inherited both transgenes. The level of fluorescence in each nucleus was quantified and the number of cells within each rhombomere was plotted according to the level of fluorescence (Figure 6H and I). No significant modification in the level and distribution of fluorescence was observed in r3 and r5 upon HS and the distribution was found unimodal in these rhombomeres (Figure 6K and L; bimodality tests: $P > 0.1$ for r3 and r5 at 35 °C and 37 °C, respectively). In contrast, in even-numbered rhombomeres the distribution of fluorescence revealed the existence of two peaks (Figure 6H–J; bimodality tests: r2 35 °C: $P < 0.05$, 37 °C: $P < 0.01$; r4 35 °C: $P < 0.05$, 37 °C: $P < 0.01$; r6 35 °C: $P < 0.001$, 37 °C: $P < 0.01$). Upon 37 °C HS, the two peaks are displaced toward higher levels of fluorescence than at 35 °C (Figure 6H–J). We interpret the highest fluorescence peak (peak 2 in Figure 6H) as corresponding to the population of cells that have stably engaged into the autoregulatory process. In contrast, peak 1

corresponds to cells that have failed to do so, although mKrox20 has transiently activated *mcherry* expression, explaining why the level of fluorescence is not null. We confirmed these interpretations using numerical simulations. For this purpose, the mathematical model was modified to introduce the *Tg(cA:h2b-mcherry)* transgene, and to take into account the large stability of the mCherry protein (in contrast to Krox20; Nadine Peyriéras, personal communication) and the possible different accessibilities of element A in its transgenic form versus endogenous element A. With these modifications, the model accounts for the experimental cell distributions, including the displacement of the peaks upon increasing initiation levels and the non-zero position of peak 1 (Supplementary Figure S9). The stability of the mCherry also explains its reduced level in even- as compared with odd-numbered rhombomeres, where expression is initiated earlier.

In conclusion, our experimental results confirm that the positive feedback loop transforms a transient pulse of Krox20 into a bimodal cell distribution in the hindbrain. As expression of *Krox20* determines cell fate, this feature is an essential function of the autoregulatory loop.

## Rhombomere size is buffered against fluctuations in the initiation signal

Simulations show that the fraction of cells that commit to the Krox20-positive fate (probability to fully activate element A) saturates when the initiation rate $\Phi_I$ is higher than the value $\Phi_I \approx 0.3\Phi_A$ (Supplementary Figure S7B). In WT embryos, we estimated $\Phi_I \approx 0.7\Phi_A$ (see section 2.5 in Supplementary Information), suggesting that the initiation stimulus is saturating and that even a 50% reduction, as in the heterozygous situation, does not significantly change the cell fate distribution. Further, reductions in the initiation signal almost linearly affect the fraction of positive cells (Supplementary Figure S7B) because of the strong impact of fluctuations at low initiation levels. Hence, there is no threshold-like behaviour and the fraction of positive cells is strongly buffered against variations in the initiation signal. As the sizes of r3 and r5 reflect the fraction of cells that commit to a Krox20-positive fate, they are also protected.

We tested this model prediction experimentally. We have previously shown that Fgf signalling in the hindbrain regulates the level of *Krox20* initiation (Chomette *et al*, 2006; Labalette *et al*, 2011). To investigate the effect of variation in the initiation signal on the probability of element A activation, we quantified r3 + r5 size and *krox20* mRNA initiation levels upon variations in Fgf signalling. For this purpose, we used a drug, SU5402, which can be added to the embryo medium and acts as a specific inhibitor of FGF receptors and, therefore, prevents FGF signalling. We compared embryos treated with SU5402 with those that were mock-treated. We used $krox20^{fh227/fh227}$ embryos to measure the level of *krox20* mRNA at 5s by RT–qPCR, corresponding only to initiation, and WT siblings to estimate the area r3 + r5 at 15s (steady-state condition). Short and long treatments with SU5402 were performed, affecting FGF signalling from different initial stages during development (Figure 4E–G). Treatments led to reductions of the initiation stimulus to 29 and 23% of the control (Figure 4H) and to reductions of the r3 + r5 area to 65 and 57% of the control, respectively (Figure 4I). Simulations with reductions in $\Phi_I$ to 29 and 23% of the control value $\Phi_I = 0.13$ mRNA/min predict that the probability to fully activate element A is reduced to 74 and 62%, respectively (Figure 4J), consistent with the experimental data.

In conclusion, this analysis reveals that autoregulation turns an initiation signal into a fraction of Krox20-positive cells that determines rhombomere size and provides robustness by dampening fluctuations in the initiation.

## Destabilization of the autoregulatory loop by repressor molecules

Experimentally, we observe that *krox20* expression starts to decline with linear kinetics from 8s and is extinguished at around 25s, whereas the model predicts that the autoregulatory loop is stable once established (Figure 4B). We hypothesized that this progressive loss of expression originates from a modification in one of the parameters of the loop around 8s. We, therefore, systematically altered parameters in the model and compared simulations with experimental data: we tested modifications in $\Phi_A$, the translation rate $\phi$, the mRNA and protein degradation rates $\Psi$ and $\psi$, and the effect of masking one of the binding sites. None of these modifications reproduced the experimental data (Supplementary Figure S10). However, we find that modifying the ratio $\beta = \lambda/\mu$ induced a linear decrease in *Krox20* expression compatible with experimental data (Figure 7A). The parameter $\beta$ controls the amount of Krox20 protein necessary to maintain element A fully activated (Supplementary Figure S5B–D) and reflects the stability of the interaction of Krox20 with element A *in vivo*.

Such a change in Krox20 binding may result from either a chromatin modification or the appearance of a repressor that affects the interaction between Krox20 and element A. We have recently shown that the transcriptional repressors Nlz1 and Nlz2 can antagonize the activity of element A in zebrafish embryos (Labalette *et al*, in preparation). As *nlz1* and *nlz2* expression becomes reinforced in r3 and r5 during somitogenesis (Labalette *et al*, in preparation), these factors may modify the *Krox20* autoregulatory system to trigger the decrease observed beyond 8s. To address this issue, zebrafish embryos were co-injected with morpholinos against *nlz1* and *nlz2* mRNAs and a time course of *krox20* expression was performed by semi-quantitative fluorescent *in-situ* hybridization. This analysis shows that the knockdown of Nlz leads to a reduction in the slope of declining *krox20* expression of ~ 2.1- and 6-fold in r3 and r5, respectively (Figure 7B and C). This indicates that the Nlz factors have a major role in the destabilization of the autoregulatory loop and the loss of *Krox20* expression.

# Discussion

In this study, we demonstrate that a direct, positive autoregulatory loop is required for amplification and maintenance of the expression of Krox20. The loop relies on a *cis*-acting element containing four Krox20-binding sites, whose function was established by a mouse knockout. We develop a mathematical model that integrates the molecular characteristics of the activation process, accounts for their stochastic nature and is constrained by quantitative data obtained *in vivo* and *in vitro*. Combination of computer simulations and experimental analyses allowed us to reach a number of major conclusions: (i) the positive feedback loop underpins a bistable switch that turns a transient input into cell fate commitment; (ii) the cell distribution between the two fates is controlled by the duration and strength of the input signal, and is regulated by the expression of only a few mRNA molecules; (iii) the transient input splits one initially homogeneous population into two different homogeneous populations; (iv) binding cooperativity and synergistic activation of transcription generate a system that is reliably controlled by low input level of activator; (v) r3 and r5 size reflects the fraction of cell that commit to a Krox20-positive fate and is strongly buffered against variations in input level size; (vi) the progressive extinction of *Krox20* expression in the hindbrain involves a destabilization of the loop by repressor molecules.

This work allows to understand at a molecular level how Krox20 activation leads to an unambiguous cell fate decision and controls the sizes of the rhombomeres. These processes are critical to vertebrate hindbrain segmentation.

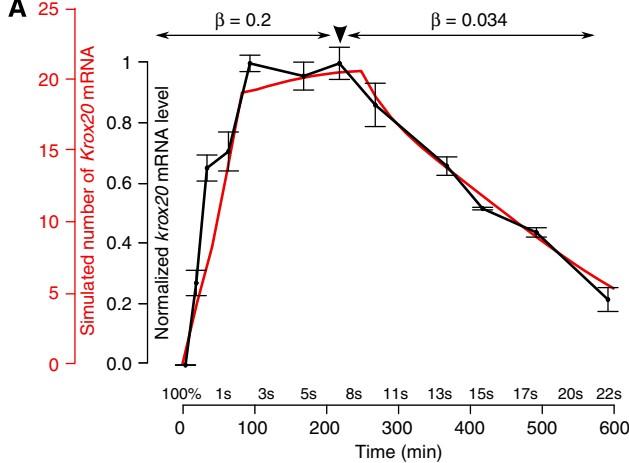

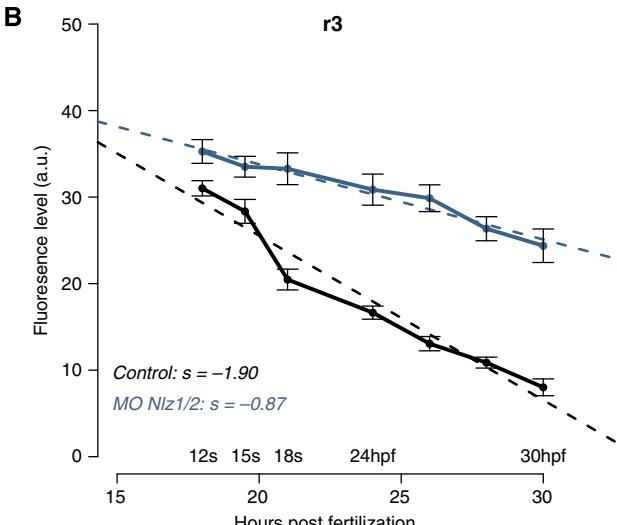

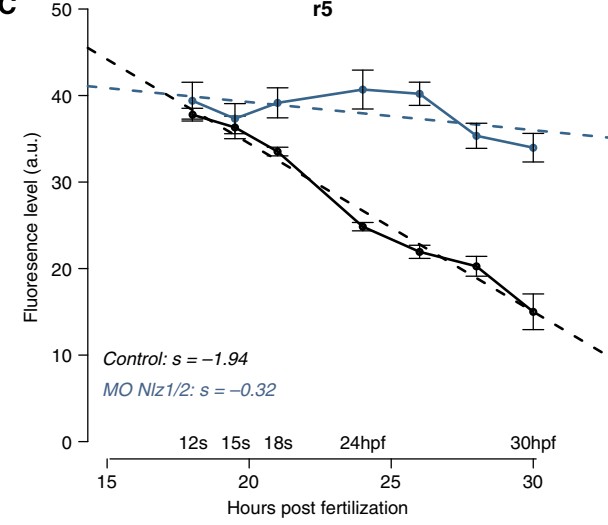

**Figure 7** Role of Nlz factors in the destabilization of the autoregulatory loop. (**A**) The experimental time course of *krox20* expression (grey) can be fully recapitulated by the model if a 6.2-fold reduction in the β-factor is introduced at $t = 250$ min (red). (**B**, **C**) Time course of *krox20* expression in r3 (**B**) and r5 (**C**) measured by fluorescent *in-situ* hybridization in embryos injected with control (grey) or *nlz1/2* morpholinos (blue). hpf, hours post fertilization; s, slope. Source data for this figure is available on the online supplementary information page.

## Autoregulation turns a transient input into a fate commitment

In mouse $Krox20^{\Delta A/\Delta A}$ and zebrafish $krox20^{fh227/fh227}$ embryos, where the autoregulatory loop is non-functional, *Krox20* expression reaches a lower plateau level and is more transient than in WT embryos, leading to an approximately fivefold reduction of the dose. This reduction results in a $2.8 \pm 0.2$-fold decrease in the number of cells positive for the marker EphA4 at 16s (Figure 1C). Furthermore, these EphA4-positive cells do not all maintain an odd-numbered identity, as their relative number, compared with WT, decreases between 10s and 16s (unpublished observations). Therefore, the absence of autoregulation leads to a significant reduction in the size of odd-numbered rhombomeres. We conclude that the positive feedback loop generates an amplification that is essential for the conversion of a transient input into a stable fate commitment.

## A bistable switch underlies hindbrain patterning

Hindbrain segmentation is characterized by homogeneous levels of *Krox20* expression within alternating Krox20-positive and -negative rhombomeres (Schneider-Maunoury *et al*, 1993; Giudicelli *et al*, 2001; Cooke and Moens, 2002). Bistability has been proposed as a means to establish such a feature (Meinhardt, 1978; Ferrell, 2002; Lopes *et al*, 2008; Zhang *et al*, 2012). Our study demonstrates that the Krox20 auto-regulatory loop provides a bistable switch generating a bimodal cell distribution with homogeneous levels of *Krox20* expression. To reach this conclusion, we developed a stochastic mathematical model based on molecular dynamics that depends on 14 parameters listed in Supplementary Table S1. We determined most of the parameter values either directly using our experimental results, or by comparing simulations with our experimental results. For two parameters, the *Krox20* mRNA production rate and the protein translation rate, we relied on the data published by Schwanhäusser *et al* (2011; 2013); see section 2.4 and 2.6 in the Supplementary Information).

Our analysis shows that the critical region where bistability occurs involves only a few mRNA molecules and that around eight molecules are already sufficient to fully activate the autoregulatory loop with a high probability and to commit cells to a Krox20-positive fate (Supplementary Figure S7I). The fully activated loop itself sustains about 23 mRNA molecules, showing that element A is strong enough to stably maintain an expression level and cell fate. The distribution of the Krox20-positive cells according to the number of Krox20 molecules depends only on the properties of element A and is independent of the characteristics of the initiation phase. This ensures that these cells form a homogeneous population. Surprisingly, for an autoregulatory system our FGF loss-of-function experiments showed that rhombomere size and, therefore, the number of Krox20-positive cells is gradually modulated by the level of the initiation signal. The model analysis revealed that this feature is entirely due to the stochasticity of the activation process at non-saturating input levels. This is in contrast with conclusions derived from a deterministic analysis of autoregulatory systems, which

predicts that all cells acquire either positive or negative fate, when the input level exceeds or falls below a threshold value, respectively. Hence, autoregulation does not only allow the specification of precisely defined developmental stages, but also its control in a graded manner.

Morphogen gradients activate target genes in a concentration-dependent manner, resulting in distinct spatial domains of expression in developing tissues (Wolpert, 1969; Meinhardt, 1978). Recently, Zhang *et al* (2012) presented a model of hindbrain patterning involving Krox20, Hoxb1a, another essential transcription factor in the segmentation process, and retinoic acid (RA), which acts as a morphogen and controls the transcription of the two transcription factors. They use coarse-grained equations to model the dynamics of *krox20* and *hoxb1a* in a single cell with cross inhibition and auto-activation. Their model predicts that the system resolves into a stripped pattern of gene expression. By adding artificial white noise to the system, they investigate the effect of fluctuations on the precision of boundary formation. By varying the noise amplitudes, they show that fluctuations in RA concentration alone induce a rough boundary, whereas additional noise in *hoxb1a/krox20* expression sharpens the boundary (see also Holcman *et al*, 2007). In the present work, we focused on Krox20 and, with our stochastic model, we dissected the molecular dynamics of Krox20 activation and derived the intrinsic fluctuations. We demonstrate that the Krox20 autoregulatory loop establishes a bistable switch and generates Krox20-positive and -negative territories. However, we find that the stochasticity of Krox20 activation precludes the formation of sharp boundaries, and additional mechanisms are therefore required for generating the sharp boundaries observed *in vivo*. The work of Zhang *et al* (2012) indicates that the cross inhibition between *krox20* and *hoxb1a* is likely to constitute a key component in the refinement of the boundaries. In a subsequent step, it would be important to develop and investigate a detailed molecular model that additionally includes *hoxb1a*. A synergetic approach that combines the methods developed in this work with the complementary model of Zhang *et al* (2012) would provide an accurate description of hindbrain patterning fully based on the molecular details.

## Robustness of rhombomere size

In the r2–r6 region, the different rhombomeres are approximately of the same AP extent and affecting their relative sizes can have deleterious consequences. Modifications in Fgf signalling have been shown to alter the relative size of r3, r4 and r5 (Marín and Charnay, 2000; Maves *et al*, 2002; Walshe *et al*, 2002; Labalette *et al*, 2011). As indicated above, endogenous *Krox20* expression occurs in saturating conditions, with an initiation level above the range required to activate the loop with high probability. This situation, combined with the graded input–output relation that we have found at lower levels of initiation, is at the origin of robustness of hindbrain patterning, as it prevents a drastic reduction in odd-numbered rhombomere size upon decreasing input signal. Indeed, we have found that rhombomere sizes are buffered from variations in the FGF-tuned initiation signal: e.g., a fourfold

reduction in the input signal results in a less than twofold reduction in the number of cells that activate the autoregulatory loop, and therefore in the size of odd-numbered rhombomeres (Figure 4E–J). This buffering effect is likely to provide robustness and have a protective role in hindbrain patterning against other genetic, environmental or stochastic sources of input variation. In this respect, we have recently shown that *Krox20* initiation is subject to a repression mechanisms distributed as a caudo-rostral gradient over the r3–r4 region, which can dramatically affect the relative size of r3 and r4 (Labalette *et al*, in preparation). The graded input–output relationship is also likely to protect the hindbrain against fluctuations of this gradient.

# Materials and methods

## Mouse and zebrafish lines

All experiments involving animals were performed in accordance with French and European regulations. The mouse $Krox20^{NA*AK}$ line was generated at the Institut Clinique de la Souris (Illkirch, France) by homologous recombination in ES cells. The $Krox20^{NA}$, $Krox20^{A*}$ and $Krox20^{AA}$ alleles were obtained as described in Figure 1A. The $Tg(cA:Krox20*_{HA})$ line was generated by transgenesis as described previously (Sham *et al*, 1993). The r2-HPAP transgenic line contains the human placental alkaline phosphatase gene under the control of an r2-specific, *Hoxa2* enhancer element (Studer *et al*, 1996). All primers used for genotyping are presented in Supplementary Table S2A. Zebrafish (*Danio rerio*) were raised and staged as described (Kimmel *et al*, 1995). The WT lines were TL and TU. The $krox20^{fh227}$ mutant line was previously described (Monk and Talbot, 2009). The $Tg(cA:h2bm-cherry)$ (gift from N Peyrieras, Gif-sur-Yvette, France) and $Tg(hsp:mKrox20_{HA})$ lines were obtained by Tol2-mediated transgenesis (Labalette *et al*, 2011).

## *In-situ* hybridization and immunohistochemistry

Mouse and zebrafish *in-situ* hybridizations were performed on whole embryos as described previously (Giudicelli *et al*, 2001) with the following digoxigenin-labelled riboprobes: *mKrox20* (Wilkinson *et al*, 1989), *mEphA4* (Gilardi-Hebenstreit *et al*, 1992) and *zkrox20* (Oxtoby and Jowett, 1993). For immunochemistry in zebrafish and chick embryos, we used rabbit anti-DsRed (1:200, Clontech) and rat anti-GFP (1:500, NacalaiTesque) as primary antibodies and Alexa594 anti-rabbit and Dy488 anti-rat (Jackson ImmunoResearch) as secondary antibodies. Nuclei were counterstained with Hoechst 33342 (Sigma).

## Morpholino injection and semi-quantitative *in-situ* hybridization

Nlz knockdown was performed by injecting four morpholinos targeting *nlz1* and *nlz2* at the one-cell stage as previously described (Hoyle *et al*, 2004). Embryos were then subjected to fluorescent *in-situ* hybridization for *krox20* using the Fastred substrate (Roche). Flat-mounted embryos were imaged on a Leica TCS sp5 confocal microscope. The level of fluorescence was then measured within a region of interest (ROI) using ImageJ on stacks of 10 sections, and was normalized by the corresponding area. The fluorescence level corresponds to the mean intensity of all pixels within the ROI (16-bit images). All embryos analysed were processed and imaged in parallel.

## Bimodality assay

Five somite $Tg(hsp:mKrox20_{HA});Tg(cA:h2b-mcherry)$ zebrafish embryos were HS treated for 10 min at 35 ($n=7$) or 37 °C ($n=8$), and allowed to develop for 4 h, until ~15s, at 28 °C in embryo medium. Anti-Cherry immunostaining was performed with rabbit anti-DsRed primary

antibodies (1:200, Clontech) and Alexa594 anti-rabbit secondary antibodies (Jackson ImmunoResearch). Nuclei were counterstained with Hoechst 33342 (Sigma). Stacks made of eleven 16-bit images of the r2–r6 region were taken with a Leica TCS sp5 confocal microscope. The $z$-step was set 1 μm, such that the total depth corresponded approximately to the depth of one nucleus. Using Fiji software, all images were first smoothened and the projections of the 11 sections were computed by summing the intensity of corresponding pixels. No background substraction, modification of contrast, luminosity or exposure was performed. Fluorescence levels were quantified in single nuclei after manual segmentation: a circular ROI of 12 pixels was placed on each nucleus identified using the Hoechst staining, and fluorescence intensity was measured and averaged over the 12 pixels. Fifty to 80 nuclei were typically quantified in each rhombomere. The plots shown in Figure 6H–L were obtained using the plot density function of the R software with the Sheather and Jones bandwidth selector. The bimodality of these plots was finally assessed statistically using the 'bimodalitytest' package in R (Holzmann and Vollmer, 2008).

## Protein extracts and gel retardation assay

Bacterial protein extracts containing Krox20 were prepared as described previously (Nardelli *et al*, 1992). For gel retardation, the DNA probes consisted of a *Hind*III–*Xho*I restriction fragment containing the chick orthologue of element A. The fragment was labelled with [$\gamma$-$^{32}$P]-labelled ATP using polynucleotide kinase (New England Biolabs). The competitors consisted of double-stranded oligonucleotides carrying Krox20-binding sites (Supplementary Table S2B). The EMSA assays were performed as described (Labalette *et al*, 2011). The proportion of DNA fragments retarded owing to complex formation was quantified using a FLA3000 PhosphoImager.

## mRNA quantitative analysis

Embryos were collected at the appropriate stage and individually dissected in embryo medium to remove yolk material and to save a piece of embryonic tissue for genotyping. They were then placed in 30 μl of the preservative solution RNA later (Ambion, Life Applied). Total RNA was isolated using the Ambion RNAqueous Micro-kit (Life Applied). Primers for *mKrox20*, *zkrox20* and the housekeeping genes *β-actin* and *eif1a* were designed in single exons to yield amplicons of ~150 bp. Reverse transcription was performed using the Superscript III enzyme (Life Invitrogen) and quantitative PCR analysis was carried out with the SYBR Green Master Mix (Roche Applied Biosystems) using a LightCycler 480 device (Roche Applied Biosystems). In each experiment, a standard curve was established by measuring the threshold-crossing cycle number (Ct) for a series of dilutions of purified genomic DNA. This allowed normalization of the assay- and primer-dependent amplification efficiency. In a single qPCR experiment, the Ct values for all genes were obtained in two duplicate reactions. Means and s.d. were calculated from a minimum of three independent experiments.

## Plasmid constructs and *in-ovo* electroporation

Various reporters were cloned into a modified pTol2 plasmid (Stedman *et al*, 2009) under the control of chick element A. The mutant versions were obtained by PCR-mediated directed mutagenesis using the high-fidelity Phusion Taq polymerase (Finnzyme). DNA sequences were verified by sequencing. Electroporation was performed as previously described (Giudicelli *et al*, 2001). Eighteen hours later, the embryos were collected in phosphate-buffered saline, fixed in 4% paraformaldehyde for 3 h and processed for immunostaining.

## Supplementary information

# Acknowledgements

We thank François Giudicelli, Patricia Gongal and Denis Thieffry for critical reading of the manuscript, and Nadine Peyriéras for personal communications and for providing us with the *Tg(cA:h2b-mcherry)* zebrafish line. This work was funded by grants from MESR, INSERM, CNRS and ANR to PC and from MESR, INSERM, CNRS and ERC to DH.

*Authors Contributions:* YXB conceived and performed a large part of the experimental work, carried out simulations and participated in the writing of the manuscript. JR developed and analysed the mathematical model, developed the simulation tools, carried out simulations and participated in the writing of the manuscript. CL participated in the experiments involving the repression of the autoregulatory loop. MAW generated zebrafish lines and initiated the project. ET participated in the analysis of the mouse mutants. CD-TD generated a mouse transgenic line. DH participated in the development of the mathematical model and in the writing of the manuscript. PG-H co-led the experimental part and participated in the writing of the manuscript. PC led the project and participated in the writing of the manuscript.

# Conflict of interest

The authors declare that they have no conflict of interest.

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
