## [Review Process File · Molecular Systems Biology]

Dissection of a Krox20 positive feedback loop driving cell fate choices in hindbrain patterning

Yassine X Bouchoucha, Jürgen Reingruber, Charlotte Labalette, Michel A Wassef, Elodie Thierion, Carole Desmarquet-Trin Dinh, David Holcman, Pascale Gilardi-Hebenstreit and Patrick Charnay

Corresponding author: Patrick Charnay, Ecole Normale Supérieure

Review timeline:	Submission date:	13 March 2013
	Editorial Decision:	18 April 2013
	Revision received:	08 July 2013
	Editorial Decision:	07 August 2013
	Revision received:	16 August 2013
	Accepted:	21 August 2013

Editors: Maria Polychronidou

Transaction Report:

1st Editorial Decision

18 April 2013

Thank you again for submitting your work to Molecular Systems Biology. We have now heard back from the three referees who agreed to evaluate your manuscript. As you will see from the reports below, the referees find the topic of your study of potential interest. They raise, however, substantial concerns on your work, which should be convincingly addressed in a revision of the manuscript.

Overall, the reviewers appreciate the presented combinatorial approach, applied for the analysis of gene regulation dynamics. However, they raise several major points, which should be carefully addressed. Among these issues are the following:

- More convincing experimental evidence needs to be provided regarding the bimodality in Krox20 expression *in vivo*.
- Several critical assumptions and parameters involved in the modeling need to be better substantiated.

Additionally, as reviewer #1 has suggested, the manuscript should be carefully re-written in order to become accessible to a broad audience.

On a more editorial level, we would like to encourage you to include the source data for figures that show essential quantitative data. (Additional information is available in the "Guide for Authors" section in our website at <http://www.nature.com/msb/authors/index.html#a3.5.2>)

If you feel you can satisfactorily deal with these points and those listed by the referees, you may wish to submit a revised version of your manuscript. Please attach a covering letter giving details of the way in which you have handled each of the points raised by the referees. A revised manuscript will be once again subject to review and you probably understand that we can give you no guarantee

at this stage that the eventual outcome will be favorable.

REFEREE REPORTS

Reviewer #1

The paper deals with dynamics of gene regulation by multiple enhancers during embryonic development. The authors use Krox20 to show that positive feedback plays a key role in maintaining the expression of the same gene induced by a transient initiation signal. Previous studies have identified that Krox20 has two enhancers, one needed for the initiation of expression and the other one responsible for maintenance. This paper provides a clear evidence that the system operates in a bistable regime and argues that positive autoregulation is needed to deal with intrinsic noise and transient induction signals. Data supporting this model are very impressive and come from experiments with both mouse and zebrafish embryos and a variety of reporter constructs. This combination of *in vivo* experiments and clean biophysical modeling is perfect for MSB. I am happy to recommend the paper for publication and have only a couple of comments.

First of all, the authors should mention the lambda system that operates on a very similar principle, with the initiation and maintenance enhancer. In fact, is there anything different between the Krox20 and lambda repressor autoregulation? In addition to two enhancers, we have multiple binding sites, etc, etc.

Second, the paper might be a hard read for the systems audience. While the techniques are quite standard in developmental genetics and gene regulation research, jumping between the experimental models, reporter constructs, conditional knockouts, etc are likely to be over head for a lot of readers interested in the dynamical systems aspects of the work. There are many ways to address this. I am sure the authors can come up with something.

Reviewer #2

This manuscript explores the autoregulation of Krox20 during transient segmentation involved in the establishment of hindbrain anterior-posterior (AP) identity. Specifically, they demonstrate that a particular enhancer element in the Krox20 locus (element A) contains Krox20 binding sites and is involved in positive autoregulation of this gene in specific rhombomeres during segmentation. Using a battery of experimental techniques- including mutant and transgenic mice, zebrafish, *in vitro* studies, and computation - to show that Krox20 binds cooperatively to sites in this enhancer element, show that loss of particular sites or the entire element significantly compromises Krox20 expression, and to argue that there is bimodality in Krox20 expression. The interaction between experiment and modeling is laudable, and the bottom line conclusions of the paper (that there is bistability in the positive feedback Krox20 loop, leading to bimodality in the Krox20 expression state among a population of cells) are conceptually attractive. There are, however, a number of questions. In particular, there appear to be internal contradictions in the formulation and assumptions involved in the stochastic modeling that need clarification. Furthermore, the experimental evidence for bimodality is not convincing.

As a minor note, the authors state in the introduction that "Fate decisions can be induced either by intrinsic cues that are asymmetrically distributed during cell division (Graham et al, 2010; Tajbakhsh et al, 2009), or extrinsic factors that are provided by the cellular environment (Briscoe, 2009)." However, these two possibilities are far from mutually exclusive. There are of course many examples of extrinsic cues inducing a fate decision that asymmetric between two daughter cells.

For the modeling, there are a few questions. It is clear that in the absence of binding elements, there is a transient Krox20 induction (Figure 2K), and the authors used this peak to estimate the value of $\tau_I=80$ min, the time at which the promoter "transitions" from an initiation (i.e. no Krox20 binding to element A) to feedback (Krox20 allowed to bind) phase. They state that transcription occurs in two phases: "a Krox20 independent initiation phase that lasts a time τ_I [and is independent of Krox20 binding], followed by an amplification/maintenance phase that relies on the Krox20 autoregulatory loop." However, it is not at all clear why - in the presence of the binding sites in element A - binding

of Krox20 at $t < t_I$ would *not* occur. As a result, the t_I "boundary" between Krox20 not being able to and then being able to contribute to Krox20 expression seems artificial and unjustified. In addition, the stated definition of t_I is confusing in light of (for example) Figure 5. These simulations do show Krox20 binding to the four binding sites in element A. Are these results intended to indicate that binding can occur, but that this binding does not increase Krox20 expression until $t=t_I$ has been reached? If so, isn't the binding state of the enhancer irrelevant before t_I ? If not, then the meaning and significance of t_I was poorly explained.

In addition, to make the analysis simpler, it was assumed that the number of Krox20 proteins was always large compared to the number of binding sites (8, counting the two loci) in element A. However, for small t (say <10 min) in Figure 5D-F the number of Krox20 proteins in every cell *is* small and comparable to the number of binding sites. Since binding is apparently allowed to occur during this time phase (as per Figure 5A-C), this violates the assumption. In addition, for $t > t_I$ in some scenarios (Figure 5D, E, G, H), there is still a population of simulated cells for which Krox20 protein copy number is low and comparable to the 8 binding sites, again violating this assumption.

For the element A mCherry reporter study, why is mCherry fluorescence observed in only half of the embryos (and why was this "expected")? Also, the experimental evidence for bimodality (Figure 6H) is not convincing. These data corresponded to 6s, which according to Figure 2K occurs at time $\sim 160-175$ min. In Figure 5D-F, by the time 175 minutes is reached, the ratio of Krox20 protein copies in the highly expressing vs. low expressing cells is at least 8-10x (~ 400 vs. ~ 30 in Figure 5D, which corresponds to the weakest initiation phase expression level considered). In contrast, Figure 6H (blue curve) shows a peak at 55-60 RFU and a minor shoulder at ~ 30 RFU (i.e. at most a 2-fold difference in expression level), and the red curve shows a peak at 20 and a minor shoulder at maybe 25. This represents all their evidence for bimodality. These shoulders could easily be experimental noise or artifact in the image processing.

The effect of fgf signaling and the use of the inhibitor is interesting, as is the discussion of the known role of fgf in rhombomere size.

In summary, this manuscript has potential to elegantly meld experimentation and computation to study interesting events in organismal development. However, there are a number of questions about the model, and more importantly the experimental evidence for bimodality in Krox20 expression in vivo is not highly convincing.

Reviewer #3

Bouchoucha et al describe a detailed analysis of the regulation of Krox20 in the vertebrate hindbrain. The authors combine molecular genetic assays of enhancer function in vivo with stochastic simulations of gene expression. They conclude that a positive auto regulation results in a bistable switch in Krox20 regulation and this allows stable expression of Krox20 in response to a transient activation. The data are also used to argue that the eventual extinction of Krox20 expression involves an interference with the auto activation loop mediated by repressors.

The regulation of Krox20 in the hindbrain is a well established system and the background to this study flows from previous work in the senior author's lab. The current study contains an appealing mix of experimental and modelling data. The quality of the data is high and some of the experiments, such as the in vivo mutational analysis and rescue of Element A (Fig 1), is well beyond the state of the art normally applied to this field. The rescue of Krox20 expression with an altered specificity protein is a strength of the study. The mathematical modelling provides a detailed framework for simulating the activity of Element A and the efforts to estimate parameters are also beyond the standard normally seen.

Nevertheless there are several issues that need to be addressed to justify and explain aspects of the study.

The principal conclusion emphasized in the Abstract - that a positive feedback loop creates bistability in Krox20 regulation - is unsurprising and the highly detailed modelling is not required for this conclusion and adds very little to it. Similarly, the title states that "stochastic cell fate

choice" underlies hindbrain patterning however the experimental data does not provide any evidence for stochasticity in the normal development of the hindbrain.

The authors do not discuss the recent work by Zhang et al (MSB 2012). This study looked in detail at the production of Krox20 and Hoxb1 stripes in the hindbrain. Zhang et al propose that cross repression between Krox20 and Hoxb1 is critical for bistability and they also use a stochastic model abstraction to simulate their data. Bouchoucha et al should compare and contrast their model with that of Zhang et al. A key question is whether the function of Hoxb1 can be ignored for the description of Krox20 regulation contained in the current study.

The authors use QPCR to assay Krox20 RNA levels in zebrafish embryos (Fig 2K). It is clear from the images that the dynamics in the two stripes in the mutant fish are different, this is not captured by the QPCR and the data should be interpreted more carefully (p7) and might indicate complications in the regulation of Krox20 that are not described in the mathematical model.

The model the authors construct relies on the accurate estimation of 14 parameters. To their credit the authors have attempted to derive these parameters from experimental data and acknowledge inevitable shortcomings in some. However, several of the parameters appear to be critical for the model and the arguments the authors develop and it is unclear if underlying assumptions are valid. For example the mRNA production rate was determined by calibration of mRNA levels in deep sequencing experiments (p6 of supplemental). The details are not provided but it appears that this is a population level analysis and does not take into account the number of cells expressing Krox20. If so this will dramatically underestimate the production rate. Moreover the cells used in the comparison are very different (neural vs fibroblasts). Independent confirmation of this parameter would greatly reassure. (Bouchoucha et al should also be aware that Schwanhauser et al recently corrected some of their absolute number data that Bouchoucha et al use to estimate rates).

Measurement of mRNA production rates and levels appears to be a critical issue for the study. The authors state that ~8 molecules are sufficient to activate the autoregulatory loop and the fully activated loop sustains ~23 mRNA. Experimental validation of this aspect of the model would strengthen the study. Moreover confidence intervals for the amounts of mRNA in the cell should be given.

In addition different systems are used to estimate different parameters (e.g. chick electroporation for mRNA production synergy and zebrafish for mRNA half life). It is plausible that these parameters (particularly lifetimes etc) will differ between species.

I was not entirely convinced that the experiments described in Fig 6 represent a reasonable test of a model prediction. First the authors examine gene expression in r2 which does not normally initiate Krox20 expression, thus it is unclear whether the same auto regulation mechanism will operate. Moreover why is r2 but not r4 or r6 analyzed? Second, the authors suggest that peak 2 of the bimodal levels of Krox20 in r2 (Fig 6H) is the population of cells that have engaged the autoregulatory loop. However the images indicate (Fig 6F,G) that the level of Krox20 in expressing cells in r2 is substantially lower than the levels of endogenous Krox20 in r3 and r5. This also raises the question of whether the mechanism is the same. It is important to note that in these experiments the authors are assaying regulation of the endogenous Krox20 alleles and it is impossible to determine whether Element A is responsible for the gene expression observed.

The authors need to demonstrate that the in ovo electroporation assay of element A (Fig 3C,D) results in the correct spatial activity of the reporter in order to validate these data.

To complement the EphA4 expression data (Fig 1C) the authors could include markers of the adjacent rhombomeres. The prediction is that they should expand to occupy the territory normally specified as r3 and r5. This would address whether there is a complete switch in identity.

1) First of all, the authors should mention the lambda system that operates on a very similar principle, with the initiation and maintenance enhancer. In fact, is there anything different between the Krox20 and lambda repressor autoregulation? In addition to two enhancers, we have multiple binding sites, etc, etc.

As suggested by the referee, we now present and discuss the example of the lambda repressor in the Introduction (p. 3).

2) Second, the paper might be a hard read for the systems audience. While the techniques are quite standard in developmental genetics and gene regulation research, jumping between the experimental models, reporter constructs, conditional knockouts, etc are likely to be over head for a lot of readers interested in the dynamical systems aspects of the work. There are many ways to address this. I am sure the authors can come up with something.

We have revised our text to make it more accessible to a larger readership. The modifications are dispersed along the text, and in particular include more detailed descriptions of the bases for the usage of different species (p. 4), the construct for Krox20* expression and the rescue experiment (p. 6), the autoregulation in zebrafish (p. 6-7), the chick electroporation assay (p. 8), the generation and analysis of double transgenic Tg(cA:h2b-mcherry; hsp:mKrox20HA) embryos (p. 13), and the presentation of and treatment with the FGF receptor inhibitor (p. 14).

Reviewer #2

1) As a minor note, the authors state in the introduction that "Fate decisions can be induced either by intrinsic cues that are asymmetrically distributed during cell division (Graham et al, 2010; Tajbakhsh et al, 2009), or extrinsic factors that are provided by the cellular environment (Briscoe, 2009)." However, these two possibilities are far from mutually exclusive. There are of course many examples of extrinsic cues inducing a fate decision that asymmetric between two daughter cells.

The sentence has been modified to take into account the comment of the referee (p. 3).

2) For the modeling, there are a few questions. It is clear that in the absence of binding elements, there is a transient Krox20 induction (Figure 2K), and the authors used this peak to estimate the value of $t_l=80$ min, the time at which the promoter "transitions" from an initiation (i.e. no Krox20 binding to element A) to feedback (Krox20 allowed to bind) phase. They state that transcription occurs in two phases: "a Krox20 independent initiation phase that lasts a time t_l [and is independent of Krox20 binding], followed by an amplification/maintenance phase that relies on the Krox20 autoregulatory loop." However, it is not at all clear why - in the presence of the binding sites in element A - binding of Krox20 at $t < t_l$ would *not* occur. As a result, the t_l "boundary" between Krox20 not being able to and then being able to contribute to Krox20 expression seems artificial and unjustified. In addition, the stated definition of t_l is confusing in light of (for example) Figure 5. These simulations do show Krox20 binding to the four binding sites in element A. Are these results intended to indicate that binding can occur, but that this binding does not increase Krox20 expression until $t=t_l$ has been reached? If so, isn't the binding state of the enhancer irrelevant before t_l ? If not, then the meaning and significance of t_l was poorly explained.

There has been a misunderstanding originating from the way we presented the evolution of the system. In our model, Krox20 can always bind to element A and autoregulation is therefore never restricted. However, in addition to autoregulation, during the so-called initiation period, a supplemental Krox20-independent production mechanism induces Krox20 expression. Thus, our definition of two phases was only based on the presence or absence of this Krox20-independent production mechanism, and as far as autoregulation is concerned, there is no difference between the two phases. To clarify the issue we have modified the legend of Figure 4A (p. 27), the model description in the main text (p. 9) and in the SI (p. 2).

3) In addition, to make the analysis simpler, it was assumed that the number of Krox20 proteins was always large compared to the number of binding sites (8, counting the two loci) in element A. However, for small t (say <10 min) in Figure 5D-F the number of Krox20 proteins in every cell *is*

small and comparable to the number of binding sites. Since binding is apparently allowed to occur during this time phase (as per Figure 5A-C), this violates the assumption. In addition, for $t > t_I$ in some scenarios (Figure 5D, E, G, H), there is still a population of simulated cells for which Krox20 protein copy number is low and comparable to the 8 binding sites, again violating this assumption.

This issue is raised because we did not sufficiently explain our methods.

First, all simulations shown in the main text were obtained by numerically solving the Master Equation 1 (eq. 1) in the SI, valid when the number of Krox20 proteins is small (1, 2,...) and comparable to the number of Krox20 binding sites. Similarly, the molecular dynamics simulations were based on eq. 1. The simulations shown in Figure 5 are therefore valid for small numbers of the Krox20 protein.

Second, to reduce complexity and start the analysis of the equation, we neglected the change in the total number of free Krox20 proteins due to binding to element A (eq. 3). We then made an additional approximation by assuming that binding to element A is a steady state (eq. 7 in the SI). Interestingly, even for small numbers of Krox20 protein, we did not observe any significant differences between simulations performed with eq. 1, eq. 3 or eq. 7, as revealed by comparison of Supplementary Figure S5B and S7J. Indeed, significant activation of element A only occurs when the number of Krox20 protein reaches a value around 40, which is large compared to the 8 binding sites. Therefore, for a low number of Krox20 proteins, autoregulation has only a small impact, explaining the similar results of simulations performed with eq. 1, 3 or 7.

To clarify these issues, we have modified the legend of Fig. 5 and the sections 1.1 and 1.3 in the SI.

4) For the element A mCherry reporter study, why is mCherry fluorescence observed in only half of the embryos (and why was this "expected")?

This remark illustrates comment 2 of the first referee and the need for us to better explain the experimental set up. The experiment is performed by crossing two transgenic lines Tg(cA:h2b-mcherry) and Tg(hsp:mKrox20HA). According to mendelian laws, half of the embryos should inherit the Tg(cA:h2b-mcherry) transgene, and express the mcherry, and only 25% both transgenes. The latter ones were of interest for the analysis. We identified them within their clutchmates after processing (heat-shock and immunochimistry) as the only embryos expressing mcherry ectopically, i.e. in r2, r4 and r6.

These points have now been clarified in the text (p. 12).

5) Also, the experimental evidence for bimodality (Figure 6H) is not convincing. These data corresponded to 6s, which according to Figure 2K occurs at time ~160-175 min. In Figure 5D-F, by the time 175 minutes is reached, the ratio of Krox20 protein copies in the highly expressing vs. low expressing cells is at least 8-10x (~400 vs. ~30 in Figure 5D, which corresponds to the weakest initiation phase expression level considered). In contrast, Figure 6H (blue curve) shows a peak at 55-60 RFU and a minor shoulder at ~30 RFU (i.e. at most a 2-fold difference in expression level), and the red curve shows a peak at 20 and a minor shoulder at maybe 25. This represents all their evidence for bimodality. These shoulders could easily be experimental noise or artifact in the image processing.

The referee questions the limited difference in fluorescence level between peaks 1 and 2 in Figure 6H, presumably corresponding to cells having stably activated and aborted the autoregulatory loop, respectively. It is true that this is a very different situation from that of the Krox20 protein presented in Figure 4D. However, we provide two types of arguments suggesting that these peaks do not simply correspond to experimental artefacts.

First, we have not presented the differences between the two situations (Figures 6H versus 4D) clearly enough to explain the modification in behaviour. Krox20 mRNA and protein are unstable (half-lives close to one hour, as measured in this work), whereas the mcherry protein is much more stable (N. Peyri ras, personal communication). Therefore, in the timeframe of our experiments, there is no degradation of mcherry and the level of fluorescence corresponds to the accumulation of the protein during the entire experiment. This explains in part why peak 1 is not close to zero as in the case of Krox20. An additional effect might come from an increased accessibility of element A to Krox20, when it is on a transgene as compared to its endogenous chromosomal position. We had incorporated these two types of modifications in the model and shown that they allowed predictions that were in close agreement with the experimental data (Supplementary Figure S8). Therefore the

data presented in Fig. 6H are consistent with the model, but reference to this part of the modelling in the main text was rather elliptic in the previous version.

Second, the analysis, previously only shown in r2 and r5, was also performed in the other rhombomeres (r3, r4 and r6). We found that the cell distribution according to the level of fluorescence was statistically bimodal in all even-numbered rhombomeres, whereas unimodal curves are observed in odd-numbered rhombomeres. This strongly argues against possible artefacts related to experimental noise or image processing.

In the present version of the manuscript we have modified Figure 6, its legend and the text (p. 13) to incorporate the data corresponding to the analysis of all rhombomeres. In addition, we briefly present the modelling of mcherry distribution in the main text to highlight the fact that the model is consistent with the experimental data (p. 13).

6) The effect of fgf signaling and the use of the inhibitor is interesting, as is the discussion of the known role of fgf in rhombomere size.

Reviewer #3

1) The principal conclusion emphasized in the Abstract - that a positive feedback loop creates bistability in Krox20 regulation - is unsurprising and the highly detailed modelling is not required for this conclusion and adds very little to it. Similarly, the title states that "stochastic cell fate choice" underlies hindbrain patterning however the experimental data does not provide any evidence for stochasticity in the normal development of the hindbrain.

The term “stochastic” was used in our study to designate the process of splitting an initial homogeneous cell population into two homogeneous groups without assuming additional spatial or temporal inhomogeneities. We did not refer to variability in the development of the hindbrain. We realize that the title might be misleading in this matter and we have therefore modified it.

2) The authors do not discuss the recent work by Zhang et al (MSB 2012). This study looked in detail at the production of Krox20 and Hoxb1 stripes in the hindbrain. Zhang et al propose that cross repression between Krox20 and Hoxb1 is critical for bistability and they also use a stochastic model abstraction to simulate their data. Bouchoucha et al should compare and contrast their model with that of Zhang et al. A key question is whether the function of Hoxb1 can be ignored for the description of Krox20 regulation contained in the current study.

We have now added a discussion of the paper of Zhang and collaborators (p. 17-18). Their model accounts for the cross-inhibition of krox20 and hoxb1a and their auto-activation, and predicts that the system resolves into a striped pattern of gene expression. Furthermore, the analysis indicates that introducing stochastic fluctuations in the RA gradient leads to boundary sharpening between the krox20-positive and -negative adjacent territories. Our present findings demonstrate that the krox20 autoregulatory loop is sufficient to establish a bistable switch and to generate krox20-positive and negative territories. However, we show that the stochasticity of krox20 expression precludes the formation of sharp boundaries. The work of Zhang and collaborators indicates that the cross-inhibition between krox20 and hoxb1a, together with the stochastic variations in morphogen level, are likely to constitute a key component in the refinement of the boundaries. Therefore, the combination of both studies provides significant, additional insights in the mechanisms of hindbrain pattern formation.

3) The authors use QPCR to assay Krox20 RNA levels in zebrafish embryos (Fig 2K). It is clear from the images that the dynamics in the two stripes in the mutant fish are different, this is not captured by the QPCR and the data should be interpreted more carefully (p7) and might indicate complications in the regulation of Krox20 that are not described in the mathematical model.

To take into account the comment of the referee, we have introduced an additional paragraph on p.10 and a Supplementary Figure (Figure S4) in which we tested the effect of introducing two shifted inputs, corresponding to initiation in r3 and r5, in the model. This allowed a fit with the experimental data very similar to the one obtained upon simulating a single “mean” input. For simplicity, we therefore maintained all following simulations with a single input.

4) *The model the authors construct relies on the accurate estimation of 14 parameter. To their credit the authors have attempted to derive these parameters from experimental data and acknowledge inevitable short comings in some. However, several of the parameters appear to be critical for the model and the arguments the authors develop and it is unclear if underlying assumptions are valid. For example the mRNA production rate was determined by calibration of mRNA levels in deep sequencing experiments (p6 of supplemental). The details are not provided but it appears that this is a population level analysis and does not take into account the number of cells expression Krox20. If so this will dramatically underestimate the production rate. Moreover the cells used in the comparison are very different (neural vs fiboblasts). Independent confirmation of this parameter would greatly reassure. (Bouchoucha et al should also be aware that Schwanhausser et al recently corrected some of their absolute number data that Bouchoucha et al use to estimate rates).*

The calibration of mRNA levels by deep sequencing was normalized for the total number of expressing cells: the sequencing was performed on mRNA extracted from dissected hindbrains, we considered that one third of the cells expressed Krox20, given that r3 and r5 represent approximately one third of the dissected region, and we normalized the number of Krox20 mRNA molecules per cell using the absolute numbers of mRNAs of six ubiquitous genes from the Schwanhaüsser study (section 2.4 of SI). We agree that this latter study has been performed on fibroblasts. However, as it concerns the expression of ubiquitous genes, we do not expect considerable variations from one cell type to another. In agreement with this idea, we found that the relative values for the six ubiquitous genes used as reference were internally consistent between the fibroblasts and the hindbrain cells. The corrigendum by Schwanhaüsser and colleagues indeed affects the estimation of the krox20 translation rate (but not the transcription rates). It was recalculated (section 2.6 of SI) and implemented into the model. The simulations were modified in consequence. This does not affect any of our conclusions.

5) *Measurement of mRNA production rates and levels appears to be a critical issue for the study. The authors state that ~8 molecules are sufficient to activate the autoregulatory loop and the fully activated loop sustains ~23 mRNA. Experimental validation of this aspect of the model would strengthen the study. Moreover confidence intervals for the amounts of mRNA in the cell should be given.*

Direct experimental validation of these numbers would require to directly measure the number of mRNA molecules in a cell within the embryo. This is not yet feasible in vertebrate embryos. We had therefore to rely on indirect estimations. The number of molecules sufficient to activate the autoregulatory loop depends on a number of parameters (mRNA lifetime, production rate of Krox20 protein from each mRNA, binding dynamics, protein lifetime and promoter strength). The protein production rate was chosen as equal to the median translation rate of 43 zinc finger transcription factors, which is reasonable and the best we can do, but still not fully satisfactory, we agree. In these conditions, it is not possible to determine a confidence interval for the amounts of mRNA per cell. We have introduced a sentence on p. 17 to make this point clear when we discuss the number of molecules.

6) *In addition different systems are used to estimate different parameters (e.g. chick electroporation for mRNA production synergy and zebrafish for mRNA half-life). It is plausible that these parameters (particularly lifetimes etc) will differ between species.*

It is true that some of the parameters were derived from different species. However, most of them were obtained from the zebrafish, in particular the half-lives. Chick electroporation experiments allowed us to determine the synergy between bound Krox20 proteins and mouse cells were used to evaluate levels of mRNA (necessary to obtain FA) and the translation rate (a parameter discussed above). Nevertheless, we think that these approximations are justified by the extreme conservation of the autoregulatory process. We have evidence that the mouse, chick and zebrafish A elements behave similarly in mouse, chick and zebrafish (Chomette et al., 2006 and our unpublished data). Furthermore, we show in this work that the chick element A can participate, together with the mouse protein, in the construction of an artificial autoregulatory loop that effectively rescues a defective mouse loop in which element A is mutated. Finally, the properties of the ectopic mouse protein in the zebrafish are fully consistent with the predictions of the model developed on the basis of the endogenous protein (Figure 4C,D).

7) *I was not entirely convinced that the experiments described in Fig 6 represent a reasonable test of a model prediction. First the authors examine gene expression in r2 which does not normally initiate Krox20 expression, thus it is unclear whether the same auto regulation mechanism will operate. Moreover why is r2 but not r4 or r6 analyzed? Second, the authors suggest that peak 2 of the bimodal levels of Krox20 in r2 (Fig 6H) is the population of cells that have engaged the auto regulatory loop. However the images indicate (Fig 6F,G) that the level of Krox20 in expressing cells in r2 is substantially lower than the levels of endogenous Krox20 in r3 and r5. This also raises the question of whether the mechanism is the same. It is important to note that in these experiments the authors are assaying regulation of the endogenous Krox20 alleles and it is impossible to determine whether Element A is responsible for the gene expression observed.*

- As indicated in the response to comment 5 of referee #2, we have incorporated the analysis of the other rhombomeres in the revised version and the data are fully consistent with those of r2 and r3.

- It is true that the levels of cherry expression reached in the even-numbered rhombomeres are much lower than those observed in r3 and r5. However, in these experiments the heat shock has been performed at 5s. As discussed in the response to comment 5 of referee #2, the cherry protein is stable. This generates a significant difference between odd- and even-numbered rhombomeres, because we are looking at the accumulation of the protein since the beginning of its expression, which occurs at different times in the two types of rhombomere. As indicated in the response to comment 5 of referee #2, modelling of the mcherry is now presented in the main text (p.13) and is fully consistent with the observed data. A sentence has been introduced to indicate the basis for the difference in mcherry levels in odd- and even-numbered rhombomeres.

- So far we have no evidence for differences in the mechanisms of the autoregulatory loop in odd- and even-numbered rhombomeres, apart from the presence of Nlz, which is not homogeneously distributed along the hindbrain (Labalette et al., manuscript in preparation). Our model supports this interpretation, since it appears to account for the experimental data in both types of rhombomere.

- We are not sure to understand the last sentence of the referee, since in these experiments we are not looking at the endogenous gene, but at an A:cherry reporter.

8) *The authors need to demonstrate that the in ovo electroporation assay of element A (Fig 3C,D) results in the correct spatial activity of the reporter in order to validate these data.*

We have previously reported the r3/r5 specificity of reporter expression when placed under the control of chick or mouse element A in chick electroporation experiments (Chomette et al., 2006, Development, 133, 1253-62). We also provide a Figure for referee perusal showing the restriction of GFP expression in r3 and r5 in such an experiment (Figure 1 for referees). We have added a sentence on p. 8 to indicate the spatial specificity of this assay.

9) *To complement the EphA4 expression data (Fig 1C) the authors could include markers of the adjacent rhombomeres. The prediction is that they should expand to occupy the territory normally specified as r3 and r5. This would address whether there is a complete switch in identity.*

As suggested by the referee, the reduction, but persistence of r3 was confirmed by direct analysis of the r2 and r4 territories, using an alkaline phosphatase reporter transgene specifically expressed in r2 (Studer et al., 1996, Nature, 384, 630-634) and in situ hybridization against Hoxb1 to reveal r4. In addition, an increase in r4 size was observed at embryonic day 9, suggesting that lack of Krox20 autoregulation may lead to re-specification of cells normally fated to belong to odd-numbered rhombomeres.

The novel data, with independent labelling of r2 and r4, are presented in p. 5-6 and in Figure 1E.

2nd Editorial Decision

07 August 2013

Thank you again for submitting your work to Molecular Systems Biology. We have now heard back from the two referees who agreed to evaluate your manuscript. As you will see from the reports below, the reviewers are cautiously supportive and express a number of concerns, which we would ask you to carefully address in a revision of the manuscript. One fundamental issue was raised by Reviewer #2, regarding the experimental data presented in Figure 6. We would like to ask you to provide the necessary quantitative information demonstrating that the observed bimodality is not

due to image processing artifacts.
Thank you for submitting this paper to Molecular Systems Biology.

REFeree REPORTS

Reviewer #2

The authors have addressed concerns with model formulation and other questions.

The real concern for this reviewer has been the quality of the experimental data in Figure 6. The two apparent peaks in the even rhombomeres upon heat shock still differ by only a factor of 2, and their existence could be due to image processing artifact, high background in some images (e.g. Fig. 6F), and/or noisy data (the data are smoothed, and the actual number of embryos analyzed, number of 1 micron stacks counted in each embryo, total number of cells counted across all embryos and stacks, and process for smoothing the data are not specified, at least in the main text, legend, and methods section). In addition, there are large differences in fluorescence between even and odd rhombomeres in the images, particularly in the non-heat shocked case. However, the even (Figs. 6H, I, J) and odd (Figs. 6K, L) blue curves are not substantially different (<2-fold different in average fluorescence levels). This raises further concerns with image analysis (e.g. image segmentation, background subtraction, and other potential artifacts). Furthermore, while high mCherry stability could lead to high background, there are other low background methods available to provide more definitive data (e.g. mRNA FISH). Currently, this reviewer still does not find the data convincing.

Reviewer #3

The revisions and additions made by the authors address the majority of technical questions I raised during the initial review. I remain of the opinion that the experimental quality of the work is very high and the data clearly support the authors' conclusions. I am still not convinced, however, that the mathematical modelling adds significantly to the work. The authors do not appear to have a clear justification for using the rather complicated stochastic model they describe. The main conclusion of the study is that a positive feedback loop creates bistability in Krox20 regulation this is demonstrated conclusively by the data, what does the model add to this conclusion?

The authors have added a brief discussion of the recently published work of Zhang et al (MSB 2012) in which cross repression between Krox20 and Hoxb1 is proposed to be critical for bistability, but it is still unclear to me how the two studies relate to one another. It would greatly strengthen the reach of the current paper if the data of Zhang et al were better incorporated into the interpretation.

2nd Revision - authors' response

16 August 2013

Reviewer #2

The authors have addressed concerns with model formulation and other questions.

The real concern for this reviewer has been the quality of the experimental data in Figure 6. The two apparent peaks in the even rhombomeres upon heat shock still differ by only a factor of 2, and their existence could be due to image processing artifact, high background in some images (e.g. Fig. 6F), and/or noisy data (the data are smoothed, and the actual number of embryos analyzed, number of 1 micron stacks counted in each embryo, total number of cells counted across all embryos and stacks, and process for smoothing the data are not specified, at least in the main text, legend, and methods section). In addition, there are large differences in fluorescence between even and odd rhombomeres in the images, particularly in the non-heat shocked case. However, the even (Figs. 6H, I, J) and odd (Figs. 6K, L) blue curves are not substantially different (<2-fold different in average fluorescence levels). This raises further concerns with image analysis (e.g. image segmentation, background subtraction, and other potential artifacts). Furthermore, while high mCherry stability

could lead to high background, there are other low background methods available to provide more definitive data (e.g. mRNA FISH). Currently, this reviewer still does not find the data convincing.

1) Concerning the limited difference in the levels of fluorescence between the two peaks in the even rhombomeres, we have provided a likely explanation by performing the simulations presented in Supplemental Figure S9. Stability of the mCherry protein and differences in accessibility between endogenous and transgenic element A are sufficient to lead to an accumulation of mCherry in absence of autoregulation (peak 1) and to fully account for a limited difference between the levels of fluorescence of the two peaks.

To make this point clearer, we have modified the legend of Fig. S9 to insist on the links between the experimental data and the simulations.

2) To alleviate the concerns of the referee about possible image processing artefacts, we now provide a detailed description of our procedure in the Materials and Methods section (p. 21), including number of embryos, stacks and cells. In particular, the entire r2-r6 region was treated as a single image, all embryos were pictured using the same parameters in non-saturating conditions, nuclei segmentation was performed manually and no background subtraction, modification in contrast, luminosity or exposure was performed. This procedure, together with the number of embryos (7-8 of each type) and the number of nuclei analysed within each rhombomere (50-80), should prevent image processing artefacts. To assess the bimodality of each curve, a statistical test was applied to the density plots, using the `bimodalitytest` package of R.

3) Concerning the difference in the level of fluorescence between odd- and even-numbered rhombomeres, we agree with the referee that the apparent large differences in the images contrasted with the limited difference in the curves. This was not due to any manipulation in image processing to generate the curves, as indicated above, but rather to an increase in contrast in Fig. B-G designed to better visualize the difference between the two cell populations in even-numbered rhombomeres. In addition, we unfortunately masked a large part of r6, where the heat shock-induced fluorescence is highest, with the insets for high magnification. We have now re-introduced the native pictures that are consistent with the curves and reorganized the figures so that the insets do not mask r6. Finally, we did not understand the comment of the referee concerning the non-heat shocked embryos, since in this case we precisely expect a large difference in fluorescence level between odd- and even-numbered rhombomeres, and moreover we did not perform any quantitative analysis.

In conclusion, we think that with the detailed information added in the Materials and Methods section, together with the modifications introduced in Figure 6, we provide convincing experimental data for bimodality, without having to rely on additional procedures that may not be well established in the zebrafish model.

Reviewer #3

The revisions and additions made by the authors address the majority of technical questions I raised during the initial review. I remain of the opinion that the experimental quality of the work is very high and the data clearly support the authors' conclusions. I am still not convinced, however, that the mathematical modelling adds significantly to the work. The authors do not appear to have a clear justification for using the rather complicated stochastic model they describe. The main conclusion of the study is that a positive feedback loop creates bistability in Krox20 regulation this is demonstrated conclusively by the data, what does the model add to this conclusion?

The purpose of our modeling is not only to draw the general conclusion that a positive feedback loop generates bistability. We use the model to integrate various experimental results and obtain the value of key fundamental parameters involved in promoter regulation, mRNA and protein dynamics. This approach provides a detailed quantitative description of the molecular mechanisms governing Krox20 activation at a cellular level. Molecular processes are driven by random events, requiring a stochastic approach to model them. Here the number of molecules (*krox20* mRNA) involved is small and thus we developed a stochastic model based on the Master equation to study the dynamics and the propagation of small fluctuations across the scales: from DNA to protein level. In addition,

in our heat-shock experiments, we obtain a graded response by applying a spatially uniform and transient input signal, which cannot be explained by a classical deterministic approach without adding ad hoc assumptions that are not derived from biological considerations. In contrast, our stochastic approach provides sufficient explanations to understand how an initial homogenous population of cells turns a transient input signal into a bimodal distribution of cell fates.

We modified the last paragraph of the introduction (p. 4) to emphasize why the use of a stochastic model is essential.

The authors have added a brief discussion of the recently published work of Zhang et al (MSB 2012) in which cross repression between Krox20 and Hoxb1 is proposed to be critical for bistability, but it is still unclear to me how the two studies relate to one another. It would greatly strengthen the reach of the current paper if the data of Zhang et al were better incorporated into the interpretation.

The work by Zhang and collaborators illustrates that morphogen gradients activate target genes in a concentration dependent manner resulting in distinct spatial domains of expression in developing tissues, while we focus here on the intracellular dynamics of the Krox20 activation process in a single cell. Indeed, Zhang et al presented a model where *hoxb1a* and *krox20* expression is induced by a spatial gradient of retinoic acid (RA). To study hindbrain patterning and the precision of boundary formation between rhombomeres, they used a reaction-diffusion equation for the spatial dynamics of RA together with coarse-grained equations for the dynamics of the *krox20* and *hoxb1a* concentrations in a single cell with autoregulation and cross-inhibition. Zhang et al incorporate fluctuations in *krox20*, *hoxb1a* and RA concentrations by an additive white noise, which is not derived from molecular dynamics. Their study focuses on the effect of noise on the precision of boundary formation. Interestingly, they find that fluctuations in RA concentration alone induce a rough boundary, but additional noise in *hoxb1a/krox20* expression sharpens the boundary.

In contrast, we investigate the molecular dynamics of Krox20 activation in a single cell by taking into account promoter regulation via several Krox20 binding sites, mRNA and protein dynamics and the effect of a transient initiation process regulated by Fgf signaling. We find that the duration and strength of the initiation process determines the probability that a cell eventually expresses *krox20*. For an initiation process that depends on the spatial position along the hindbrain, this results in a spatial distribution of *krox20*-positive and negative cells. The work of Zhang et al suggests that cross-inhibition between *krox20* and *hoxb1a* is a key feature for the refinement of rhombomere boundaries. We have not addressed the precision of boundary formation as we only focused on *krox20* dynamics. However, our stochastic approach gives access to the intrinsic noise in *krox20* activation, which was found to be important for boundary refinement. In a next step it would be interesting to study the molecular dynamics of cell fate choice by including the effect of *hoxb1a*.

We conclude that our work addresses different questions than the one of Zhang and collaborators and both should be seen as complementary. A synergetic approach would consist in merging methods of these two papers so that our model serves as an input to the model of Zhang et al. The result would be an accurate description of hindbrain patterning based on the molecular details.

In the present version of the manuscript, we have modified and extended the discussion on p. 17-18 to incorporate these ideas.